# *Lhx3/4* initiates a cardiopharyngeal-specific transcriptional program in response to widespread FGF signaling

C. J. Pickett⬢, Hannah N. Gruner, Bradley Davidson⬢*

Department of Biology, Swarthmore College, Swarthmore, Pennsylvania, United States of America

* bdavids1@swarthmore.edu

**Data Availability Statement:** Raw and processed RNA sequencing data available from NCBI Gene Expression Omnibus (accession number GSE250325).

## Abstract

Individual signaling pathways, such as fibroblast growth factors (FGFs), can regulate a plethora of inductive events. According to current paradigms, signal-dependent transcription factors (TFs), such as FGF/MapK-activated Ets family factors, partner with lineage-determining factors to achieve regulatory specificity. However, many aspects of this model have not been rigorously investigated. One key question relates to whether lineage-determining factors dictate lineage-specific responses to inductive signals or facilitate these responses in collaboration with other inputs. We utilize the chordate model *Ciona robusta* to investigate mechanisms generating lineage-specific induction. Previous studies in *C. robusta* have shown that cardiopharyngeal progenitor cells are specified through the combined activity of FGF-activated *Ets1/2.b* and an inferred ATTA-binding transcriptional cofactor. Here, we show that the homeobox TF *Lhx3/4* serves as the lineage-determining TF that dictates cardiopharyngeal-specific transcription in response to pleiotropic FGF signaling. Targeted knockdown of *Lhx3/4* leads to loss of cardiopharyngeal gene expression. Strikingly, ectopic expression of *Lhx3/4* in a neuroectodermal lineage subject to FGF-dependent specification leads to ectopic cardiopharyngeal gene expression in this lineage. Furthermore, ectopic *Lhx3/4* expression disrupts neural plate morphogenesis, generating aberrant cell behaviors associated with execution of incompatible morphogenetic programs. Based on these findings, we propose that combinatorial regulation by signal-dependent and lineage-determinant factors represents a generalizable, previously uncategorized regulatory subcircuit we term "cofactor-dependent induction." Integration of this subcircuit into theoretical models will facilitate accurate predictions regarding the impact of gene regulatory network rewiring on evolutionary diversification and disease ontogeny.

## Introduction

### Overview of developmental gene regulatory networks

Developmental gene regulatory networks (GRNs) program spatiotemporal gene expression, ultimately determining embryonic cell fate decisions and morphogenesis [1–4]. The

**Funding:** This work was supported by two grants received by BD. The American Heart Association grant number 20AIREA35080013 (https://professional.heart.org/en/research-programs/aha-funding-opportunities) along with the National Science Foundation grant number 8077804 (https://www.nsf.gov/). The funders had no role in study design, data collection and analysis, decision to publish, or preparation of the manuscript.

**Competing interests:** The authors have declared that no competing interests exist.

**Abbreviations:** ANP, anterior neural plate; ASMF, atrial siphon muscle founder; ASV, anterior sensory vesicle; CPF, cardiopharyngeal founder; CPP, cardiopharyngeal progenitor; EGF, epidermal growth factor; FFL, feed-forward loop; FGF, fibroblast growth factor; FGFR, FGF receptor; GRN, gene regulatory network; HP, heart precursor; HPF, hours postfertilization; LSA, lineage-specific coactivator; LSR, lineage-specific repressor; MapK, Map kinase; MG, motor ganglion; NE, neuroectodermal; PBS, phosphate-buffered saline; RTK, receptor tyrosine kinase; TF, transcription factor; VEGF, vascular/endothelial growth factor.

cardiogenic GRN in the invertebrate chordate *Ciona robusta* provides a useful overview of one such network (Fig 1). During early stages of development, network circuits composed of genes encoding transcription or signaling factors dictate changes in the regulatory state or signaling status of each cell lineage (Fig 1A–1C). As development proceeds, these early circuits feed into effector circuits driving transient changes in cell behavior or more stable changes in cell identity associated with differentiation (Fig 1D–1F). Some portions of developmental GRNs encode cell autonomous programs in which inherited regulatory states dictate cascading shifts in gene expression (Fig 1A). However, substantial portions of developmental GRNs encode non-cell-autonomous programs, in which signals regulate the cell fate of neighboring lineages through modifications of signal-dependent transcription factor (TF) activity (Fig 1B). Previous studies have elucidated many fundamental features of GRNs, including recurrent network motifs [5], the bistability of cell fate [6], modular subcircuits [7,8], along with linkages between regulatory and effector circuits [3,9]. However, critical aspects of developmental GRN structure and function remain poorly described. One major gap involves integration of microenvironmental, extrinsic cues, including paracrine signals and matrix factors. How are subcircuits structured to permit robust responses to inherently noisy extrinsic cues? How do GRNs process a limited set of signal-dependent inputs to generate a vast array of differential transcriptional outputs? Addressing these gaps is required to productively investigate GRN rewiring associated with evolution and disease progression.

## The hierarchical topology of developmental GRNs

Developmental GRNs consist of semi-independent, hierarchical modules or circuits. Distinct populations of progenitor or precursor cells are programed by distinct territorial gene networks. Each territorial GRN can be divided into regulatory modules, or subcircuits, that execute discrete, cascading functions as first elucidated in reference to echinoderm skeletogenic mesoderm specification [10]. Network motifs, such as feedback or feedforward loops, have also been considered to act as regulatory modules [11]. It has been proposed that there may be a limited set of recurring subcircuits or motifs that are deployed for related functions across both developmental and physiological GRNs [8,11–14]. Because theories regarding network motifs were derived from studies of bacterial signaling, extrinsic inputs are well integrated. In contrast, initial theories regarding developmental subcircuits were largely derived from knockdown of lineage-specific TFs as opposed to highly pleiotropic signal-dependent TFs. Thus extrinsic inputs have not been fully integrated into current theoretical overviews of developmental subcircuits.

## The pleiotropic input of signals in developmental GRNs

Signals play an integral role in coordinating and refining the deployment of nearly all territorial GRNs [15]. Remarkably, this rich array of developmental functions largely involves only 10 signaling pathway families [15–17]. Although these families have diversified to generate numerous paralogous signaling components, each family still regulates transcription through a limited set of signal-dependent TFs, each of which binds a similar, family-specific binding site motif. This paradigm is well illustrated by the receptor tyrosine kinases (RTKs), a signaling pathway family that encompasses many key developmental signals, including epidermal growth factors (EGFs), vascular/endothelial growth factors (VEGFs), and fibroblast growth factors (FGFs). All of these signals mediate changes in transcription through a shared set of transduction cascades that includes the Map kinase (MapK) pathway. MapKs, regardless of the upstream signal that drives their activity, often regulate transcription through phosphorylation of Ets family TFs [18,19] (Fig 1B). All Ets factors bind a highly stereotyped motif (GGAW)

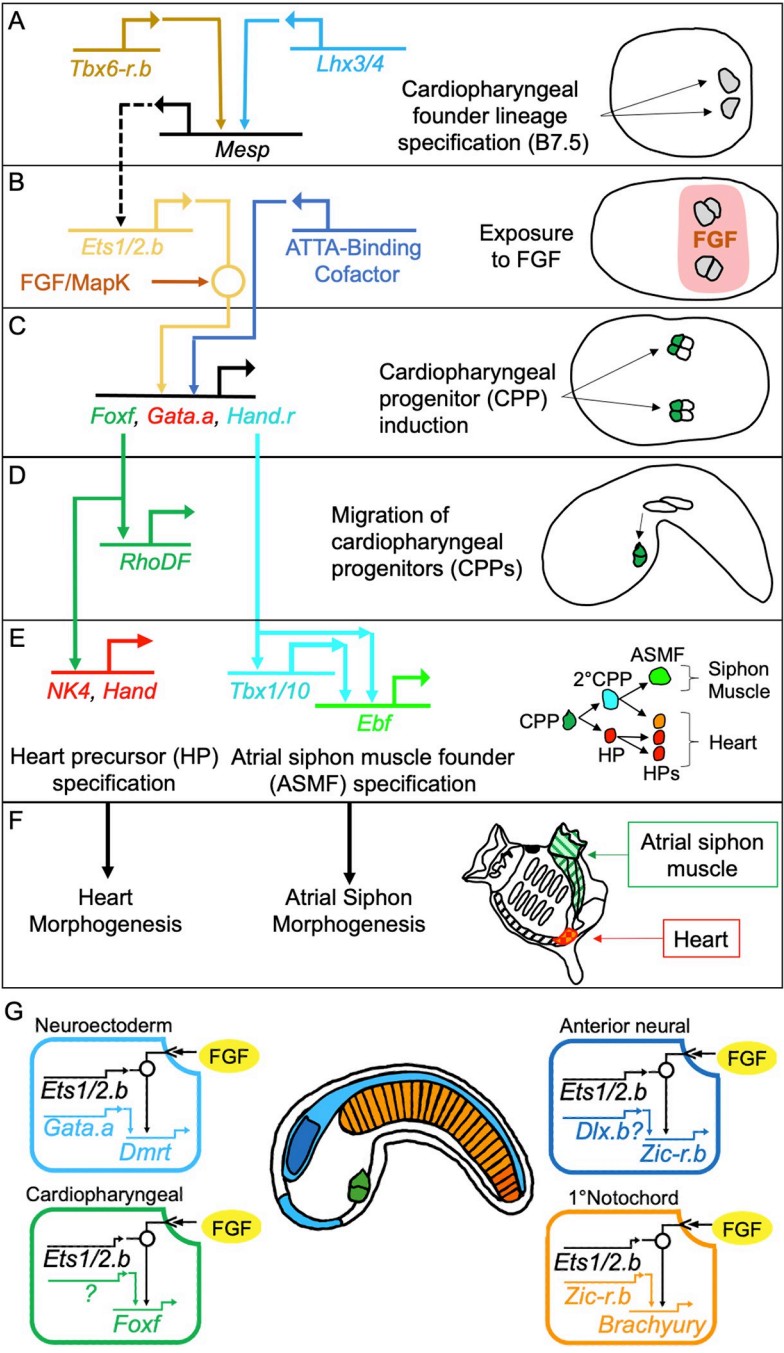

**Fig 1. *Ciona robusta* cardiopharyngeal specification and development.** Panels pair regulatory circuits **(A-E)** or summaries of morphogenetic events **(F)** with corresponding schematics. Red and orange HPs represent primary and secondary heart precursors respectively. **(G)** Illustration of *C. robusta* regulatory networks involving FGF/Ets and known or putative lineage-determining cofactors. Colors used in network diagrams pair with shaded territories in cartoon tailbud *C. robusta* embryo. ASMF, atrial siphon muscle founder; CPP, 2°CPP, primary and secondary cardiopharyngeal progenitors; FGF, fibroblast growth factor; HP, heart precursor; MapK, Map kinase.

[20,21]. The resulting tendency for diverse, widely deployed signaling pathway families to regulate transcription through a single, shared motif represents a conundrum. How does the genome encode such a vast array of temporally and spatially distinct transcriptional responses

to functionally indistinguishable inputs? Lineage-specific transcriptional responses to pleiotropic signaling are thought to be driven by differential deployment of cofactors, referred to as lineage-determining TFs [22,23]. However, despite the central role of Ets family factors and other signal-dependent TFs in developmental patterning, the transcriptional partners that mediate discrete transcriptional outputs remain poorly characterized [24–26]. More broadly, the mechanisms by which these partners determine signal-dependent transcriptional outputs have not been characterized and relevant subcircuits have not been incorporated into theoretical models [8]. We aim to address these fundamental gaps in inductive specificity through in-depth analysis of the *C. robusta* cardiopharyngeal GRN.

## Elucidation of gene regulatory networks in *Ciona robusta*

*Ciona robusta* (also referred to as *Ciona intestinalis Type A* [27,28]) provides an excellent platform for investigating developmental GRN structure and function. *C. robusta* is a tunicate, a clade of invertebrate chordates that are the closest sister group to the vertebrates [29]. However, tunicate genomes were not subjected to 2 rounds of duplications that occurred within the vertebrate lineage. The resulting lack of paralogs for many key signaling and TFs leads to greatly reduced GRN complexity. For example, there is a single gene encoding the FGF receptor (FGFR) in *C. robusta* versus at least 4 distinct *FGFR* genes in most vertebrate genomes. Additionally, the *C. robusta* genome is extremely compact. Relatively short intergenic regions permit rapid identification of *C. robusta* regulatory elements, and the ability to easily generate large numbers of transgenic *C. robusta* embryos has facilitated in-depth characterization of these elements [30]. These advantages have been productively exploited to generate a high-resolution map of the *C. robusta* cardiopharyngeal GRN [31,32].

## The *Ciona robusta* cardiopharyngeal gene network

In *C. robusta* embryos, 2 founder cells (the cardiopharyngeal founders or CPFs) give rise to the heart and a subset of pharyngeal muscle cells. According to current models (Fig 1), CPFs are specified by the 110-cell stage when overlapping expression domains of *Lhx3/4* and *Tbx6-r.b* activate *Mesp* exclusively in these 2 cells [33,34] (Fig 1A). *Mesp* is presumed to activate expression of *Ets1/2.b* [35–37], making the CPF cells competent to respond to FGF signaling (Fig 1B). During gastrulation, CPFs are exposed to uniform FGF signaling, but due to differential FGFR trafficking, only a subset of CPF daughter cells activate the MapK cascade required to execute the downstream cardiopharyngeal progenitor (CPP) specification program (green cells in Fig 1C) [36,38]. In these daughter cells, MapK-dependent phosphorylation of Ets1/2.b initiates progenitor induction. Activated Ets1/2.b partners with an ATTA-binding cofactor to drive transcription of CPP genes, including *Foxf*, *Hand.r*, and *Gata.a* (Fig 1C) [39]. *Foxf* initiates a migratory program by driving transcription of effector genes, including *RhoDF* (Fig 1D) [37,40]. During early tailbud stages, pairs of bilateral CPPs migrate along the ventral trunk epidermis and eventually meet along the ventral midline (Fig 1D) [36,41]. These progenitors then undergo a series of asymmetrical divisions to produce heart precursor (HP) and atrial siphon muscle founder (ASMF) lineages (Fig 1E). Each of these lineages execute distinct GRNs eventually driving distinct morphogenetic programs in the juvenile (Fig 1F) [42–45].

## Pleiotropic roles for the FGF/MapK/Ets pathway in *C. robusta*

*C. robusta*'s cellular simplicity has facilitated characterization of a wide array of FGF-dependent inductive events [46]. Indeed, FGF plays a predominant role in *C. robusta* embryonic patterning. During early cleavage stages, FGF secreted from the central endoderm progenitors specifies notochord and neural plate progenitor lineages, along with a range of "mesenchymal"

lineages that give rise to postlarval mesodermal tissues. As mentioned above, FGF also specifies the CPP lineage during this early embryonic period. In later stages, FGF subspecifies a number of distinct lineages within the neural plate, CPPs, ectoderm, and endoderm [36,47–49]. As with CPP induction, many of these FGF-dependent specification events have been shown to alter transcription through the MapK/Ets pathway [47,48]. According to current models, shared reliance on FGF is translated into differential specification through deployment of distinct, lineage-determining Ets cofactors, some of which have been identified. For example, *Gata.a* and *Zic-r.b* serve as the Ets cofactors for neuroectodermal and primary notochord specification, respectively (Fig 1G) [18,50,51]. However, many of these presumptive partner factors, including the ATTA-binding factor driving Ets-dependent expression of CPP genes, have not been characterized.

Here, we show that the homeobox TF *Lhx3/4* serves as the *Ets1/2.b* partner responsible for coactivation of the transcriptional program driving *C. robusta* CPP specification. Because *Lhx3/4* also participates in CPF lineage specification (Fig 1A) [34], it effectively regulates CPP gene expression through a multistep coherent feed-forward loop. Strikingly, we find that misexpression of *Lhx3/4* in the anterior neural plate (ANP) lineage is sufficient to suppress the native FGF/MapK-dependent ANP program and initiate ectopic activation of the CPP specification program [18,48,52]. However, *Lhx3/4* misexpression also initiates ectopic activation of a neuroectodermal motor ganglion (MG) program. The resulting execution of incompatible morphogenetic programs appears to drive highly aberrant cell behaviors. Based on these findings, we propose that lineage-determining cofactors play an instructive role, dictating rather than merely facilitating lineage-specific transcriptional responses to pleiotropic signals. We also argue that integration of subcircuits delineating the contribution of lineage-determining factors is essential for accurate modeling of developmental GRNs and the impact of GRN rewiring on evolutionary diversification and disease ontogeny.

## Results

### *Lhx3/4* is required for cardiopharyngeal progenitor cell migration

Previous studies have demonstrated that phosphorylated Ets1/2.b works in conjunction with an unknown ATTA-binding cofactor to initiate CPP gene expression (Fig 1B, 1C and 1G) [39]. Because ATTA resembles the core homeobox TF binding motif [53,54], we hypothesized that the CPP cofactor is a homeobox factor. The *C. robusta* genome contains 86 homeobox TFs [55]. However, a previous study revealed that only a small subset of these factors are enriched in sorted CPF lineage cells that were subjected to expression analysis just prior to CPP specification (Fig 1A and 1B) [39]. We employed CRISPR-Cas9 to separately knock down 4 of these candidate cofactors (*Lhx3/4*, *Lmx1*, *Pax3/7*, and *Pou2*; Fig 2A). Varying combinations of 2 single guide RNAs (sgRNAs) targeting each of these candidates were expressed ubiquitously using the *C. robusta* U6 promoter [56] (see S1 Table for details of all sgRNAs utilized). To ensure targeted knockdown in the CPF lineage, embryos were cotransfected with *Mesp>nls::Cas9::nls* in which the *Mesp* enhancer is deployed to drive robust and lineage-specific knockdown in CPF lineage cells [35]. Embryos were also cotransfected with *Mesp>H2B::GFP*, allowing us to determine whether knockdown of any of these candidate genes blocks CPP specification as reflected by disrupted CPP migration from the tail into the trunk (Figs 1C, 1D and 2C) [35]. In the negative control group (electroporated with *Mesp>H2B::GFP*, *Mesp>nls::Cas9::nls*, and an empty sgRNA vector (sgControl)), an average of 10.1% of embryos displayed abnormal migration (Fig 2B). This baseline rate accords with other published results [37] and likely reflects generalized developmental defects resulting from dechorionation and transgenic expression of *GFP* or *Cas9*. We also performed a positive

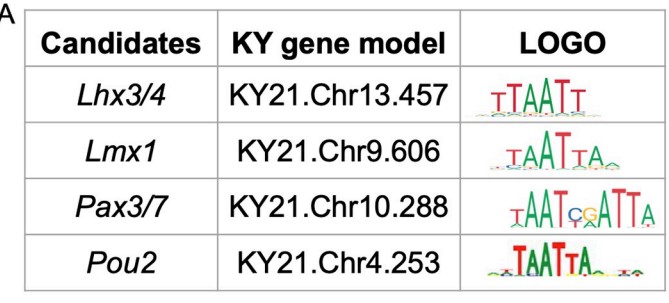

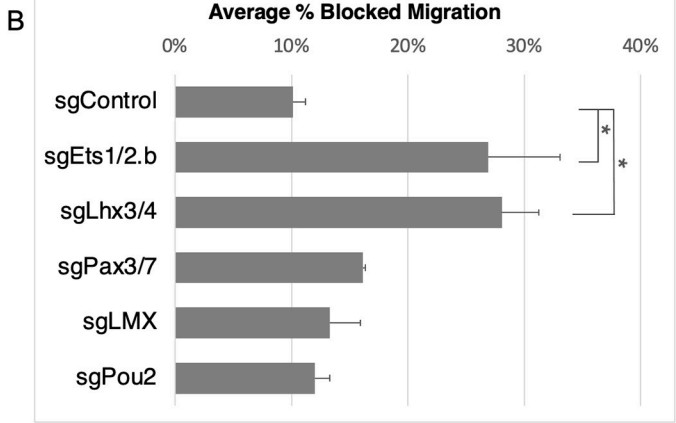

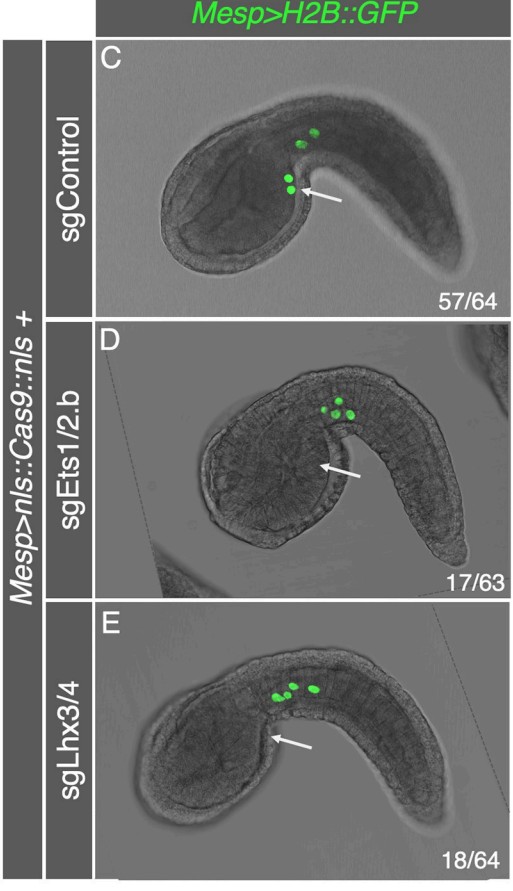

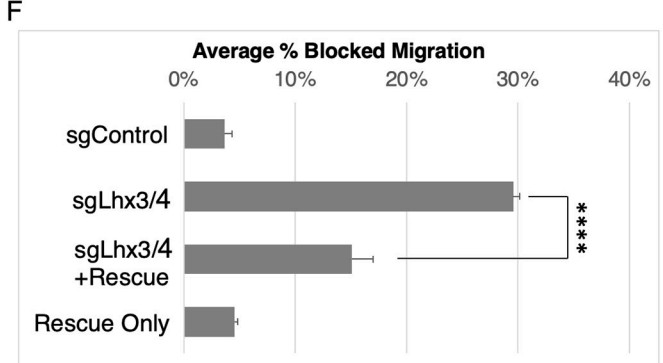

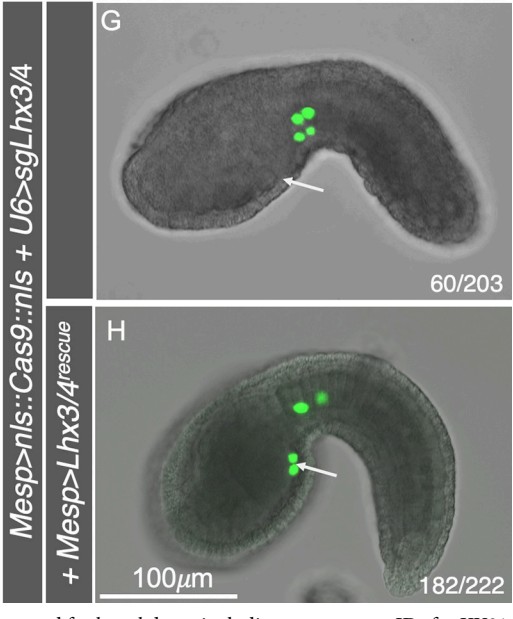

**Fig 2. *Lhx3/4* is required for CPP migration. (A)** Table displaying candidate genes targeted for knockdown including names, gene IDs for KY21 gene models [57], and DNA-binding LOGOs [58–60]. LOGOs have been loosely aligned to highlight shared ATTA (TAAT) motifs. See http://ghost.zool.kyoto-u.ac.jp/default_ht.html [57] for conversion of KY21 IDs into the previous generation KH2012 IDs. **(B)** Graph displaying the average percent of embryos exhibiting blocked migration over 2 trials. Negative control group (10.1%, $n = 78$). *Ets1/2.b*-targeting positive control group (26.9%, $n = 63$, $p = .0097$). *Lhx3/4* knockdown group (28.1%, $n = 64$, $p = .0061$). Underlying data can be found in S1 Appendix. **(C-E)**

Representative tailbud stage embryos electroporated with *Mesp>nls::Cas9::nls*, *Mesp>H2B::GFP*, and sgRNAs as indicated. (**F**) Graph displaying the average percent of embryos exhibiting blocked migration over 3 trials. Negative control group (3.7%, *n* = 194). *Lhx3/4*-targeting positive control group (29.6%, *n* = 203). *Lhx3/4* rescue group (15.1%, *n* = 222, *p* < .0001). *Lhx3/4* rescue only control group (4.5%, *n* = 223). Underlying data can be found in S1 Appendix. (**G, H**) Representative tailbud stage embryos electroporated with *Mesp>nls::Cas9::nls*, *Mesp>H2B::GFP*, *U6>sgLhx3/4*, and *Mesp>Lhx3/4^{rescue}* as indicated. (**C-E, G, H**) In these panels, white arrows mark normal CPP position at this stage of development. Green is nuclear GFP fluorescence. Numbers indicate quantity of embryos displaying phenotype depicted by each image over total.

control, using a pair of sgRNAs to knockdown *Ets1/2.b*. This manipulation led to a modest but significant suppression of CPP migration (26.9% +/− 6.1%) in comparison to the negative control group (Fig 2B and 2D). The limited penetrance of this phenotype aligns with previous studies in which *Mesp* was deployed as the *Cas9* driver and may reflect the narrow time frame for *Cas9*-driven gene editing, which must occur between expression of *Cas9* protein (at approximately 5 hours postfertilization (HPF)) and CPP specification (at 6.5 HPF). Among the 4 homeobox candidates, targeted knockdown of *Lhx3/4* using a combination of sgRNAs (sgLhx3/4^{185} and sgLhx3/4^{886}, subsequently referred to as sgLhx3/4) was the only manipulation that resulted in a significant suppression of CPP migration (Figs 2B, 2E and S1). The level of suppressed migration in these samples was similar to that seen in the positive control group (sgEts1/2.b). Although these results indicate that *Lhx3/4* is required for CPP migration, we conducted a number of follow-up experiments to buttress this interpretation.

To evaluate predicted gene editing of *Lhx3/4* by the sgLhx3/4^{185/886} pair of sgRNAs, we sequenced the *Lhx3/4* locus in transfected CPP cells (S2 Fig). Because the *Mesp* regulatory element only drives *Cas9* in a small percentage of embryonic cells, transgenic tailbud stage embryos (*Mesp>H2B::GFP*, *Mesp>nls::Cas9::nls*, *U6>sgLhx3/4*) were dissociated and GFP-labeled CPP-lineage cells were FACS collected. These sorted cells were then used as a PCR template to sequence the *Lhx3/4* locus. *Lhx3/4* sgRNAs were designed to ensure that alternatively spliced forms of *Lhx3/4* [61] would be silenced by deletion of an approximately 6,000-bp region between the first and third exon of the longer isoform (S2A Fig). Sequencing of PCR amplicons derived from sorted cell gDNA confirmed the predicted deletion in the *Lhx3/4* locus (S2B Fig). Because we did not conduct further analysis of sgRNA knockdown for the other targets, we cannot rule out a potential role for one of these alternative candidate factors in CPP specification.

To determine whether the observed impact of the *Lhx3/4* sgRNAs on CPP migration specifically reflected *Lhx3/4* knockdown instead of off-target effects, we co-expressed the sgLhx3/4 constructs along with an sgLhx3/4-protected form of *Lhx3/4* in the CPPs (*Mesp>Lhx3/4^{rescue}*) and examined the impact on CPP migration (labeled with *Mesp>H2B::GFP*). Reflecting the initial screen, targeted *Lhx3/4* knockdown blocked migration in 29.6% of transfected embryos (averaged across 3 trials; Fig 2F and 2G). In contrast, coexpression of *Mesp>Lhx3/4^{rescue}* in matched samples significantly reduced the incidence of blocked migration (15.1% across 3 trials; Figs 2F, 2H and S3A). Notably, knockdown of *Lhx3/4* did not decrease expression of the *Mesp* reporter (*Mesp>H2B::GFP*), indicating knockdown of *Lhx3/4* did not interfere with initial CPF specification (S3B Fig). Taken together, these results demonstrate that *Lhx3/4* is required for CPP migration and support the hypothesis that *Lhx3/4* serves as the cofactor for CPP specification.

### *Lhx3/4* is required for *Hand.r* expression in cardiopharyngeal progenitors

To further test the hypothesized role for *Lhx3/4* in CPP specification, we examined the impact of *Lhx3/4* knockdown on *Hand.r* expression. *Hand.r* is expressed in bilateral pairs of CPPs immediately following their specification (early neurula; Fig 3A, black arrowheads). At this stage, *Hand.r* is also expressed in a bilateral set of anterior cells (Fig 3A, white arrowheads;

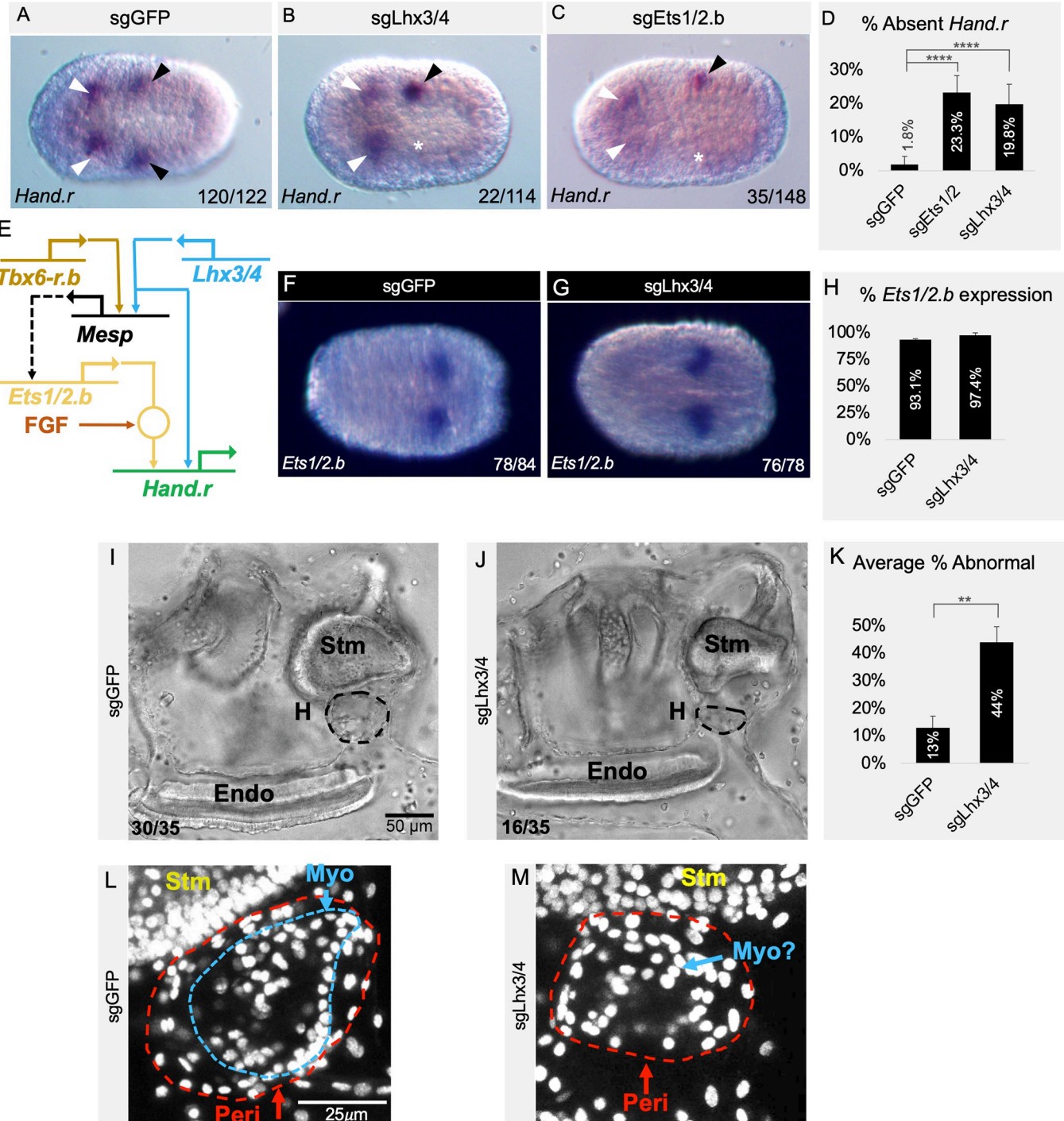

**Fig 3. *Lhx3/4* is required for *Hand.r* expression and heart formation. (A-C, F, G)** Representative neurula stage embryos coelectroporated with *Mesp>nls::Cas9::nls* and sgRNAs as indicated. (**A**) Control embryo displays wild-type *Hand.r* expression pattern in the CPPs (black arrowheads) and anterior lineages (white arrowheads). (**B, C**) sgRNAs targeting *Lhx3/4* (**B**) or *Ets1/2.b* (**C**) often resulted in embryos that are missing *Hand.r* expression in the CPPs on one side of the embryo (as indicated by a white asterisk). (**D**) Graph showing the average percent of CPPs lacking *Hand.r* expression across 2 trials. Control group *n* = 122 (1.8%); *Ets1/2.b* knockdown *n* = 148 (23.3%), *p* < .00001; *Lhx3/4* knockdown *n* = 114 (19.8%), *p* < .00001. Underlying data can be found in S1 Appendix. (**E**) GRN showing hypothesized role of *Lhx3/4* (see text). (**F, G**) Representative embryos probed for *Ets1/2.b* expression. (**H**) Graph displaying two-trial average percentage of CPPs expressing *Ets1/2.b*, for control group *n* = 84 (93.1%) and for *Lhx3/4* knockdown group, *n* = 78 (97.4%), *p* = .17896. All embryos positioned anterior to the left. Numbers indicate quantity of embryos displaying phenotype depicted in each image over total. Underlying data can be found in S1 Appendix. (**I, J**) Representative transmitted light images taken from movies of juveniles coelectroporated with *Mesp>nls::Cas9::nls* and sgRNAs as indicated (3I is from S1 Movie and 3J is from S2 Movie). Black dotted circle indicates the heart. (**K**) Quantification of the percent of heart defects observed over

2 trials, for controls (*n* = 35, 13%) and for *Lhx3/4* knockdown samples (*n* = 35, 44%), *p* = .0041). Underlying data can be found in S1 Appendix. **(L, M)** Representative confocal Z-stacks of juvenile hearts electroporated with *Mesp>nls::Cas9::nls* and sgRNAs as indicated. Nuclei stained with DAPI (white). **(L)** Control hearts show stereotypical anatomical structures including a myocardial tube (Myo, outlined in blue) and pericardial sphere (Peri, outlined in red), adjacent to the stomach (Stm). **(M)** Representative *Lhx3/4* knockdown heart illustrating one of the observed abnormal phenotypes similar to that shown in S3 Movie. Heart morphology is grossly abnormal and putative myocardial cells are disorganized and hard to distinguish. Note that another small, grossly malformed heart was visible on the other side of the stomach in this sample, but this was difficult to visualize using a projected Z-stack.

[62]). In approximately 98% of control embryos (*Mesp>nls::Cas9::nls, U6>sgGFP*), *Hand.r* exhibited a wild-type expression pattern (Fig 3A and 3D). As predicted by our hypothesis, knockdown of *Lhx3/4* (*Mesp>nls::Cas9::nls, U6>sgLhx3/4*) led to a consistent and significant reduction in the percentage of CPPs displaying bilateral *Hand.r* expression (Fig 3B and 3D). A similar result was seen in positive controls (sgEts1/2.b; Fig 3C and 3D). In both *Lhx3/4* and *Ets1/2.b* knockdowns, embryos often showed unilateral loss of *Hand.r* expression (white asterisks, Fig 3B and 3C), consistent with mosaic, unilateral incorporation of transgenes often observed in electroporated *C. robusta* embryos [63,64].

## Knockdown of *Lhx3/4* does not impact *Ets1/2.b* expression in the CPPs

Based on our knockdown results, we hypothesized that *Lhx3/4* has 2 critical roles in CPP specification (Fig 3E). During early cleavage stages, *Lhx3/4* serves as a cofactor for cell-autonomous specification of the CPF lineage [34]. Subsequently, during gastrulation, *Lhx3/4* serves as the *Ets1/2.b* cofactor mediating signal-dependent specification of the CPP lineage. Although our results demonstrate that *Lhx3/4* is required for *Hand.r* expression and CPP migration, it is not clear whether this is due to the hypothesized role of *Lhx3/4* in CPP specification or reflects an earlier, previously characterized role of *Lhx3/4* in CPF lineage specification (Fig 3E). In particular, if *Mesp>nls::Cas9::nls*-dependent knockdown of *Lhx3/4* disrupted *Mesp* expression, this may lead to loss of *Ets1/2.b* expression, indirectly blocking signal-dependent specification of CPPs. Although it is unlikely that the *Mesp>nls::Cas9::nls* would knockdown *Lhx3/4* early enough to disrupt founder lineage specification, we addressed this concern by examining *Ets1/2.b* expression. In control trials (*Mesp>nls::Cas9::nls* and *U6>sgGFP*), normal, bilateral *Ets1/2.b* expression was consistently observed in 93.1% of transgenic embryos (Fig 3F and 3H). Knockdown of *Lhx3/4* generated a similar impact on *Ets1/2.b* expression (Fig 3G and 3H). These results provide further support for the hypothesis that *Lhx3/4* serves as the *Ets1/2.b* cofactor during signal-dependent CPP specification.

## *Lhx3/4* knockdown disrupts heart formation

According to our model, knockdown of *Lhx3/4* and the subsequent loss of CPP specification should disrupt the formation of a beating heart, which is first observed after metamorphosis in 4-day old juveniles. To test this prediction, we reared *Lhx3/4*-knockdown and control animals to an early juvenile stage (Stage 6: Days 7 to 9; [65]). We first examined gross heart morphology using low-power imaging of living juveniles and then fixed them for higher-resolution confocal analysis. At this stage of juvenile development, the heart is composed of an outer pericardial sphere surrounding a single-cell layer myocardial tube that undergoes regular peristaltic contractions (Fig 3I and 3L and S1 Movie). In control trials, an average of 13% of animals exhibited mildly abnormal phenotypes, largely consisting of abnormally slow peristaltic contraction (Fig 3K). As predicted by our model, *Lhx3/4* knockdown led to a significant increase in the percentage of juveniles displaying phenotypically abnormal hearts. In these experimental trials, 44% of *Lhx3/4*-knockdown animals exhibited a range of morphological defects including frequent gross abnormalities (Fig 3J, 3K and 3M). In some of these samples, the

heart was composed of an empty pericardial cavity with no discernible myocardium (Fig 3J and S2 Movie). In other samples, heart size was reduced and the myocardium was severely disorganized (Fig 3M and S3 Movie). These results align with defects in CPP migration and unilateral loss of CPP gene expression observed in *Lhx3/4*-knockdown embryos (Figs 2E and 3B).

## Misexpression of *Lhx3/4* is sufficient for ectopic activation of CPP enhancers in the anterior neural plate

As detailed in the introduction, shared reliance on an inductive signal, such as FGF, is thought to drive differential specification by deployment of distinct, lineage-determining cofactors. This model predicts that shifting the expression of an Ets cofactor from one FGF-dependent lineage to another should drive ectopic expression of genes normally restricted to the first lineage. Due to widespread deployment of FGF/MapK/Ets-dependent induction in early embryos (Fig 1G), *C. robusta* provides a valuable experimental platform for testing this prediction. In particular, we focused on determining whether ectopic expression of the CPP Ets cofactor *Lhx3/4* would be sufficient to drive expression of CPP genes in the ANP, which also undergoes FGF-dependent induction. As illustrated in Fig 4A, *Dlx.b* serves as the presumptive cofactor for FGF/Ets-dependent expression of ANP genes, including *Zic-r.b* [66]. FGF-dependent induction of the ANP occurs during the 64-cell stage in a row of cells that express both *Dmrt* and *Dlx.b* (Fig 4B). By the neurula stage, this row has undergone 2 divisions and 2 of the resulting 4 rows of ANP cells express *Zic-r.b* (expression of this gene is repressed in the anterior 2 rows, which are fated to form the palps) [48]. A similar cofactor-dependent induction circuit also drives CPP specification, except in this case *Lhx3/4* serves as the cofactor (Fig 4C). Thus, by the early neurula stage, FGF-dependent induction has resulted in differential expression of lineage-specific TFs (including *Zic-r.b* or *Foxf*) in both ANP and CPP lineages (Fig 4B).

We used the *Dmrt* enhancer [48] to ectopically express the putative CPP cofactor *Lhx3/4* in ANP lineage cells at the 64-cell stage. Because the FGF/MapK/Ets1/2.b pathway induces ANP lineage gene expression at this stage (Fig 4A), the cofactor-dependent induction model predicts that ectopic expression of *Lhx3/4* will activate the cardiopharyngeal specification program in ANP progenitors (Fig 4D). We cotransfected embryos with either a *Foxf* or *Hand.r* reporter construct (*FoxfΔepi>LacZ or Hand.r>LacZ*) to assess activation of the CPP specification program. These constructs contain well-characterized enhancer elements with *Ets1/2.b* and *Lhx3/4* binding sites that are required for FGF-dependent reporter activity in the CPP lineage (Figs 1C and 4C) [39]. In control samples (*FoxfΔepi>LacZ* or *Hand.r>LacZ* + *Mesp>H2B::GFP*), reporter activity was almost entirely restricted to CPP lineage cells (Figs 4E, 4G, 4K, 4O and S4). Strikingly, in experimental samples (+*Dmrt>Lhx3/4*), ectopic reporter activity was observed in the ANP of nearly all *FoxfΔepi>LacZ* embryos and in the majority of *Hand.r>LacZ embryos* (arrowheads, Figs 4F, 4G, 4L, 4O and S4).

Although these results indicate that *Lhx3/4* is sufficient for FGF/MapK/Ets-dependent induction of CPP genes in the ANP lineage, it is also possible that ectopic *Lhx3/4* drives CPP gene expression independently of FGF induction. To distinguish between these alternative hypotheses, we expressed *Lhx3/4* in the ANP lineage while blocking FGF/MapK induction using U0126, a well-established inhibitor of the MapK pathway [67,68] As observed in previous experiments, misexpression of *Lhx3/4* (*Dmrt>Lhx3/4*) led to ectopic *FoxfΔepi>LacZ* or *Hand.r>LacZ* reporter activity in the neural plate of DMSO-treated embryos (Fig 4H, 4J, 4L and 4O). As predicted by the cofactor-dependent induction model, disruption of the MapK pathway completely disrupted *FoxfΔepi>LacZ* activity in both CPPs and neural plate cells (Fig 4I and 4J). Disruption of the MapK pathway also consistently and robustly disrupted ectopic *Hand.r>LacZ* activity in the majority of U0126-treated embryos (Fig 4M and 4O).

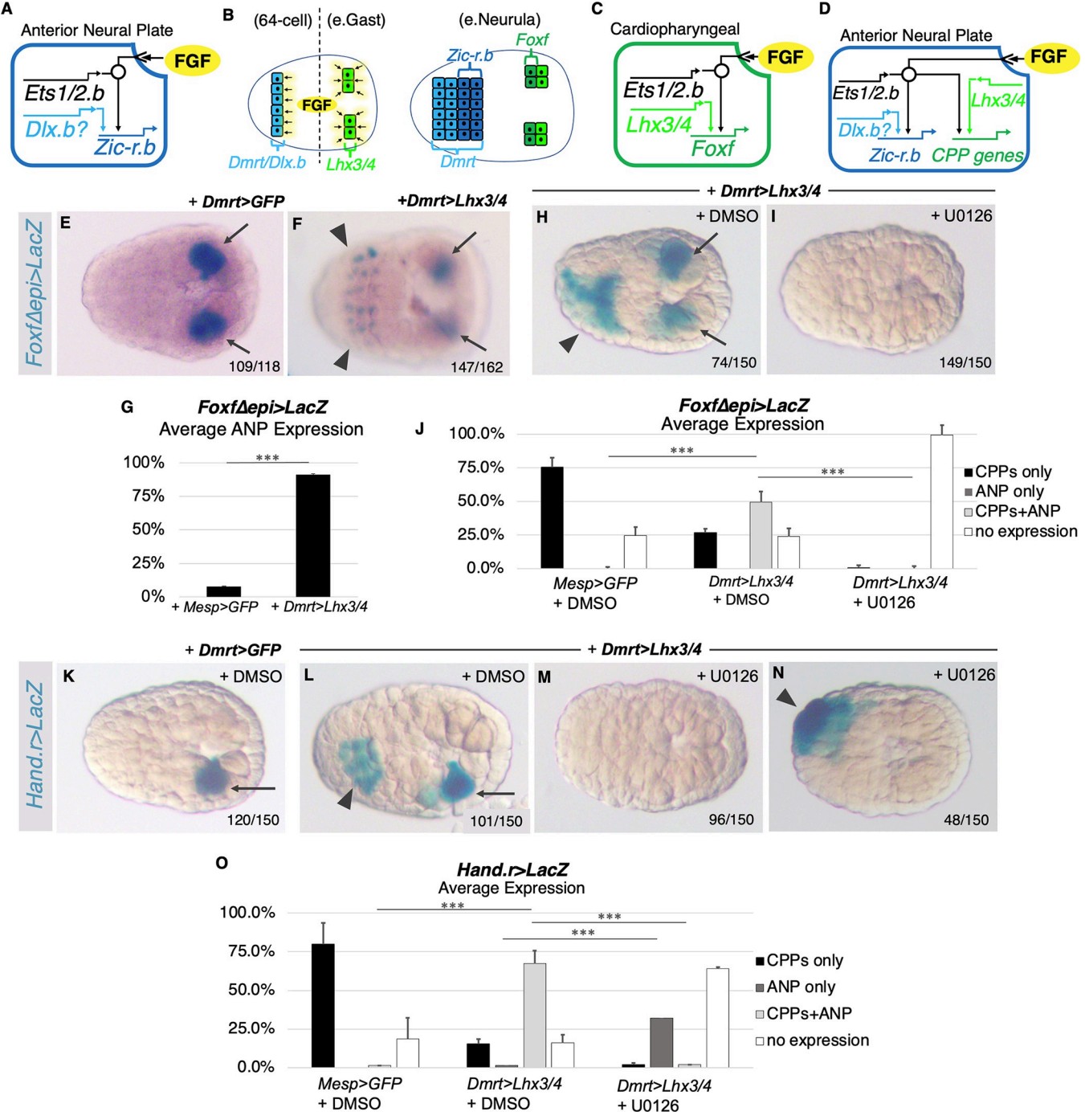

**Fig 4. Misexpression of *Lhx3/4* is sufficient for ectopic, FGF-dependent activation of CPP reporters in the anterior neural plate.** (A) Illustration of *C. robusta* regulatory circuit for ANP specification. (B) Diagram depicting induction of the ANP and CPP lineages. (C) Illustration of the regulatory circuit for CPP specification. (D) Predicted impact of ectopic cofactor misexpression (see text). (E) Representative control embryo displaying *Foxf* reporter expression in the CPP lineage (black arrows). (F) Representative embryo electroporated with *FoxfΔepi>LacZ* and *Dmrt>Lhx3/4* displaying ectopic *Foxf* reporter expression in the ANP (black arrowheads). (G) Graphical representation of results from panels E and F showing average levels of *Foxf* reporter activity in the neural plate over 2 trials. Control embryos (7.8%, *n* = 114), experimental *FoxfΔepi>LacZ* and *Dmrt>Lhx3/4* transgenic samples (91.3%, *n* = 162; *p* < .0001). Underlying data can be found in S1 Appendix. (H) Representative *FoxfΔepi>LacZ, Dmrt>Lhx3/4* transgenic embryo treated with DMSO displaying *Foxf* reporter expression in the CPF lineage (black arrows) and neural plate (black arrowhead). (I) Representative embryo electroporated with *Foxf>LacZ* and *Dmrt>Lhx3/4* treated with U0126. Note the absence of *Foxf* reporter expression in both the neural plate and CPPs. (J) Graphical representation of results from panels H and I showing average levels of *Foxf* reporter activity in the neural plate and CPPs over 3 trials. Each condition *n* = 150. Underlying data can be found in S1 Appendix. (K) Representative *Dmrt>GFP* control embryo treated with DMSO displaying *Hand.r* reporter expression in the CPP lineage (black arrows). (L)

Representative *Hand.r>LacZ, Dmrt>Lhx3/4* transgenic embryo treated with DMSO displaying *Hand.r* reporter expression in the CPP lineage (black arrow) and neural plate (black arrowhead). **(M, N)** Representative embryos electroporated with *Hand.r>LacZ* and *Dmrt>Lhx3/4* after treatment with U0126. Note that in the majority of embryos, *Hand.r* reporter expression is absent in both the ANP or CPP lineages **(M)** while some embryos retain expression in the ANP **(N, arrowhead). (O)** Graphical representation of results from panels K through N showing average levels of *Hand.r* reporter activity in the neural plate and CPPs over 3 trials. Each condition *n* = 150. Underlying data can be found in S1 Appendix. Bar and asterisks (***) in G, J, and O indicate *p*-values of less than .0001. ANP, anterior neural plate; CPF, cardiopharyngeal founder; CPP, cardiopharyngeal progenitor; FGF, fibroblast growth factor.

Intriguingly, although this treatment completely eliminated *Hand.r>LacZ* reporter activity in CPPs, about one-third of treated embryos retained reporter activity in ANP lineage cells (Fig 4N and 4O). These results indicate that differences in the regulatory circuits underlying CPP expression of *Foxf* and *Hand.r* led to differences in their reliance on the FGF/MapK/Ets pathway. More broadly, these results indicate that lineage-determining cofactors instruct, rather than facilitate, lineage-specific responses to shared signals (see Discussion).

## Ectopic expression of *Lhx3/4* disrupts anterior neural plate morphogenesis

Specification programs often activate morphogenetic modules that dictate the cell behavior of newly specified lineages [9,69]. Thus, cofactor-dependent induction of lineage-specific specification programs in response to a shared signal will also lead to differential execution of downstream morphogenetic modules. This logical corollary to the cofactor-dependent induction model predicts that ectopic, overlapping expression of 2 cofactors would result in execution of 2 likely incompatible, morphogenetic modules. We tested this prediction by examining the impact of ectopic *Lhx3/4* expression (*Dmrt>Lhx3/4*) on morphogenesis of *Dmrt>LacZ*-labeled ANP cells. Our model predicts that ectopic expression of *Lhx3/4* will activate incompatible ANP and CPP morphogenetic modules and disrupt normal cell behavior (Fig 5A). As seen in control embryos, by the mid-tailbud stage, posterior rows of ANP cells that express both *Dmrt* and *Zic-r.b* invaginate to form the anterior sensory vesicle (ASV) of the central nervous system, which lies beneath the epidermis (arrowhead, Fig 5B) while anterior rows, in which *Zic-r.b* expression is repressed, form the palps (arrow, Fig 5B). Transgenically labeled ANP cells in all control mid-tailbud stage embryos displayed these behaviors (Fig 5B and 5D). In contrast, ectopic expression of *Lhx3/4* (*Dmrt>Lhx3/4*) appeared to block invagination (Fig 5C and 5D). Transgenically labeled cells were consistently observed to lie superficial to the epidermis, forming a cluster that protruded dorsally (Fig 5C, arrowhead). Additionally, ANP lineage cells failed to spread anteriorly to form the palp primordia. These results align with our model, indicating that overlapping cofactor expression leads to execution of incompatible morphogenetic programs in response to a shared signal. In particular, disruption of ANP morphogenesis could reflect *Lhx3/4*-dependent activation of a CPP migration module (Fig 1C). Alternatively, overlapping activation of ANP and CPP morphogenetic modules might disrupt ANP morphogenesis through misregulation of cell proliferation. To begin testing these hypotheses, we examined the impact of *Lhx3/4* on ANP morphogenesis in more detail.

## Ectopic *Lhx3/4* induces precocious migratory behavior in anterior neural plate lineage cells

To begin assessing whether misexpression of *Lhx3/4* disrupted ANP morphogenesis through inappropriate activation of a CPP migration module (Fig 1C), we conducted live imaging of transgenic embryos in which ANP cells were labeled with GFP (*Dmrt>GFP*) (Fig 5E and S4 and S5 Movies). In initial tailbud stage control embryos (stage 18), *Dmrt>GFP*-labeled ANP cells have formed a well-organized, dorsal neural plate. The anterior half of the ANP is broad with a sharp, well-defined border, and it extends to the anteriormost tip of the developing

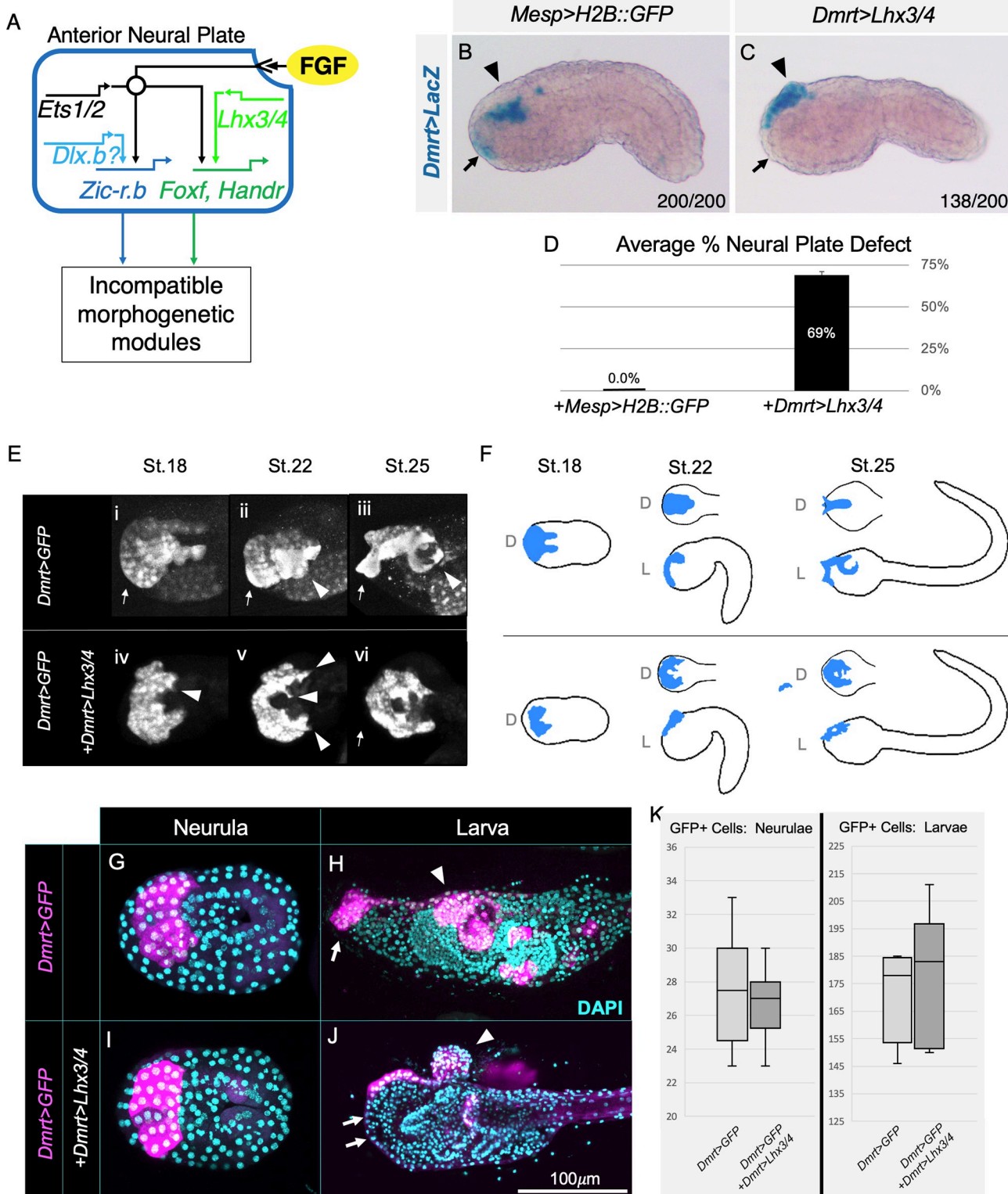

**Fig 5. Ectopic *Lhx3/4* expression causes abnormal, migratory cell behavior in the neural plate but does not alter cell numbers. (A)** Predicted activation of incompatible morphogenetic modules due to ectopic expression of the CPP lineage-determining factor *Lhx3/4*. **(B)** Representative control embryo in which *Dmrt>LacZ*-positive ANP cells are observed in the palps (arrow) and invaginated cells of the developing ASV. Note that ASV cells are at this stage covered by a layer of unstained epidermis (arrowhead). **(C)** Representative early tailbud embryo electroporated with *Dmrt>LacZ* and *Dmrt>Lhx3/4*. Note that *Dmrt>LacZ*-positive cells protrude from the anterior dorsal trunk and are not covered by epidermis (arrowhead). Staining is

also absent from the anterior tip of the embryo (arrow) where the palps normally develop. **(D)** Graph of average percent of early tailbud embryos displaying neural plate defects across 2 trials; controls (*n* = 200, 0%) and *Dmrt>Lhx3/4* samples (*n* = 200, 69%). Underlying data can be found in S1 Appendix. **(E, F)** Effect of *Lhx3/4* on neural plate morphogenesis. Stages indicated above each image or illustration column. **(E)** Still images taken from representative movies of control (i-iii; S4 Movie) and experimental *Dmrt>Lhx3/4* embryos (iv-vi; S6 Movie). See text for details. **(F)** Schematics illustrating phenotypes shown in the previous panel. D = dorsal view, L = lateral view. **(G-K)** Effect of *Lhx3/4* on cell division. Representative neurulae **(G, I)** and larvae **(H, J)** expressing transgenes as indicated on the left. Transgenic GFP expression domain shown in magenta and DAPI-labeled nuclei in cyan. **(K)** Quantification of GFP-positive (GFP+) cells from experiments represented in panels **G, I** (left plot) and panels **H, J** (right plot). Two trials, control neurulae *n* = 9 (avg. 27.6 GFP+ cells), *Dmrt>Lhx3/4* neurulae, *n* = 9 (avg. 26.8 GFP+ cells), control larvae *n* = 5 (avg. 170.8 GFP+ cells), *Dmrt>Lhx3/4* larvae *n* = 6 (avg. 178.5 GFP+ cells). Underlying data can be found in S1 Appendix. ANP, anterior neural plate; ASV, anterior sensory vesicle; CPP, cardiopharyngeal progenitor.

embryo (arrow, Fig 5Ei). The posterior half of the ANP has invaginated beneath the epidermis (Fig 5B, 5Ei and 5F) and bilateral cell columns extend from the posterior end. By the late tailbud stage (stage 22), the ANP has begun to narrow and distal cells begin to form dense placodes that will give rise to the palps (arrow, Fig 5Eii). Labeled ANP cells are well organized in regular rows with a sharp border and no visible protrusions. A pair of large cells that resemble migratory Eminens-like neurons (recently described in [70]; arrowheads, Fig 5Ei) begin to emerge from the posterior portion of the developing ASV (Fig 5Eii, arrowhead). The bilateral posterior columns have merged to form a single, well-organized narrow band. In early larvae (stage 25), the *Dmrt>GFP*-labeled cells in the ANP continue to neurulate, converging to form a narrow, medial tube. At the anterior tip, palp primordia begin to protrude and differentiate (Fig 5Eiii, arrow). In the posterior region, presumed Eminens-like neurons complete their previously characterized migratory path to form a ring around the developing sensory vesicle (Fig 5Eiii, arrowhead and S4 Movie).

As observed in the fixed samples, ectopic expression of *Lhx3/4* (*Dmrt>Lhx3/4*) severely disrupts neural plate morphogenesis (Fig 5Eiv-vi). Summary illustrations comparing typical ANP morphology in control and experimental samples are provided in Fig 5F. By the initial tailbud stage (stage 18), *Dmrt>GFP*-labeled ANP cells are already noticeably disorganized. The ANP lacks discernable rows, and the border is poorly defined. Additionally, the neural plate fails to extend to the distal palp region, and posterior bilateral columns are also not observed (Fig 5Eiv). As embryos proceed through late tailbud and early larval stages (stages 22 and 25), the ANP fails to invaginate and narrow (Fig 5Ev and 5Evi). There is also no sign of palp placode formation or morphogenesis (arrow, Fig 5Eii). In the posterior region, presumed Eminens-like neurons fail to form a ring around the developing sensory vesicle (Fig 5Eiii, arrowhead, and S6 Movie). Notably, throughout the observed stages, ANP cells appear to be highly dynamic, extending outside of the neural plate border and exhibiting signs of irregular protrusive activity (arrowheads, Fig 5Ev and S6–S8 Movies). Some cells appear to exhibit aberrant migration, and in one instance, a cell is seen to migrate from the right to the left side of the ANP cluster (S6 Movie). This increased protrusive activity, and aberrant migratory cell behaviors observed in the experimental samples suggest that misexpression of *Lhx3/4* drives inappropriate activation of a CPP migration module in the ANP lineage.

## Ectopic *Lhx3/4* does not increase proliferation in the anterior neural plate

To determine whether ectopic *Lhx3/4* might also disrupt ANP morphogenesis by altering cell proliferation, we quantified *Dmrt>GFP*-labeled cells in control and experimental samples (+*Dmrt>Lhx3/4*) fixed at early neurula (stage 14; Fig 5G and 5I) or at late larvae (24 HPF; Fig 5H and 5J). Interestingly, *Lhx3/4* had no discernible impact on ANP cell number or neural plate morphology in early neurula stage embryos (Fig 5I and 5K). Similar numbers of *Dmrt>GFP* labeled cells were arranged in a stereotypical grid at the dorsal anterior of the embryo across all samples. Thus, *Lhx3/4*-dependent disruption of ANP morphogenesis begins

in the 2-hour window between stage 14 and stage 18, when abnormal morphogenesis was observed in our live imaging studies. In line with our previous results, *Dmrt>Lhx3/4* larvae displayed severe developmental defects including an extruded dorsal ANP cluster (arrowhead, Fig 5J, compare to Fig 5H) and lack of ANP extension into the anterior, distal palp region (arrows, Fig 5J). Interestingly, we also observed that pigmentation associated with pigment cell sensory structures in the cerebral vesicle [71,72] were absent in *Dmrt>Lhx3/4* larvae (Fig 5J). Thus, *Lhx3/4* expression appears to impact some aspects of ANP subspecification. However, these defects were not accompanied by any detectable, consistent shift in cell numbers (Fig 5K). Taken together, these results suggest that *Lhx3/4*-dependent impacts on cell proliferation do not contribute to abnormal ANP morphogenesis.

## Evaluating the impact of ectopic *Lhx3/4* on gene expression in anterior neural plate cells

To more rigorously assess the impact of ectopic *Lhx3/4* on ANP gene expression and evaluate alternative hypotheses regarding the regulatory circuits underlying cofactor-dependent induction, we conducted comparative RNA-sequencing analysis. To accomplish this, *Dmrt>GFP*-labeled ANP lineage cells were collected by FACS from dissociated control embryos or experimental embryos coelectroporated with *Dmrt>Lhx3/4* (Fig 6A). For each condition, triplicate samples were used to generate mRNA libraries for sequencing. Triplicate control or experimental samples clustered together, and intergroup sample similarity was greater than between groups (Fig 6B). To begin assessing the impact of ectopic *Lhx3/4* on ANP gene expression, we first focused on genes that displayed significant (adjusted $p$-value of $< .05$) and robust changes in expression (fold change greater than or equal to 1.5, threshold log2 +/−0.58; Fig 6C). Through this analysis, we distinguished 567 up-regulated genes and 865 down-regulated genes relative to the control group (Fig 6C). To highlight changes in the ANP regulatory program, we conducted a similar analysis focused solely on annotated TFs (Fig 6D) [62]. As expected, these cells displayed robust enrichment of *Lhx3/4* expression. In line with our previous data, *Lhx3/4* misexpression led to significant and robust up-regulation of some primary targets of FGF-dependent induction in the CPP lineage including *Foxf*, *Hand.r*, and *Irxb* (homologs to human *FOXF2*, *HAND2*, and *IRX6*, respectively) [39]. However, expression of many characterized primary CPP targets were either not impacted (*Irxa*, *Soxh*, and *Gsc*) or down-regulated (*Elk1*, *Foxp*, and *SoxC*). *Foxf* is considered to be a key regulator of subsequent CCP specification and morphogenesis [37,40,73]. To investigate whether ectopic activation of *Foxf* was sufficient for activation of downstream target genes, we examined the expression of 26 characterized *Foxf* targets, as delineated through prior ATAC-seq and loss of function studies (S2 Appendix) [74]. Strikingly, the expression of a majority of these characterized targets (including *RhoDF*, *Gata.a*, *Ddr*, *SLC17A5*, *TSPAN4/9*, *TMEM150A*, and *Fzd10/9*) were not significantly impacted by this manipulation, and only 3 were up-regulated. Thus, while ectopic *Lhx3/4* is sufficient for partial activation of the primary cardiopharyngeal program in ANP cells, further execution of this program was severely abbreviated. Additionally, we observed that some of the most highly up-regulated TFs are associated with MG patterning, including *Vsx*, *Nkx6*, *Islet*, and *Pax6* [75,76]. Previous studies have demonstrated that *Lhx3/4* is expressed in a subset of MG interneuron and motoneuron precursors in *C. robusta* [61,76] and contributes to MG patterning in *Drosophila melanogaster* and vertebrate embryos [77–79]. Thus, it appears that misexpression of *Lhx3/4* was able to activate MG programs ectopically in the ANP lineage, potentially disrupting execution of both ANP and CPP programs. MG and ANP precursors originate in the neuroectoderm. Thus, MG patterning programs may be more

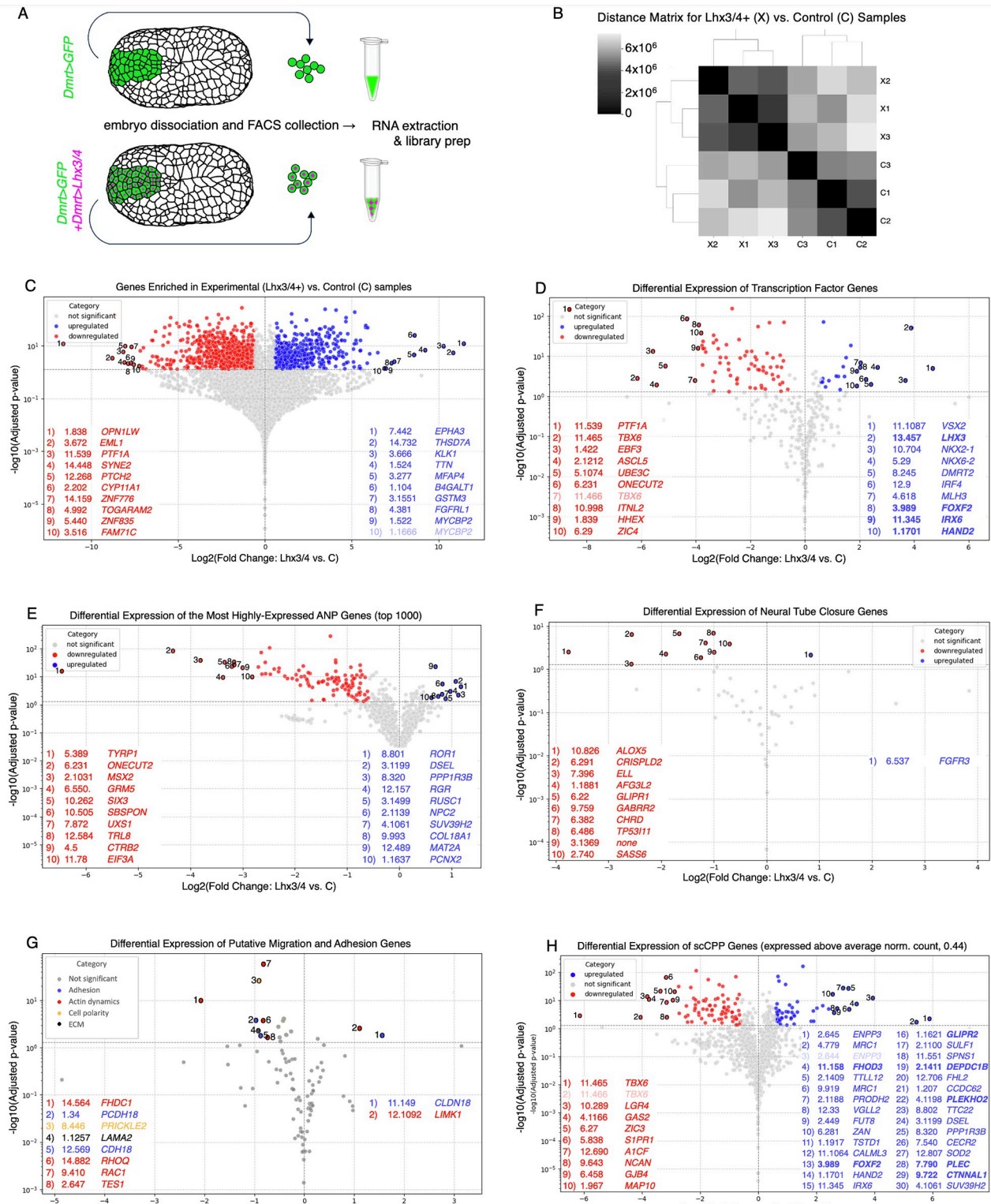

**Fig 6. Comparative RNA-seq analysis delineates the impact of ectopic *Lhx3/4* on ANP gene expression. (A)** Diagram of the experimental setup. **(B)** Sample-to-sample distance matrix. C1–C3 and X1–X3 refer to the 3 control and 3 experimental samples, respectively. **(C–H)** Volcano plots of differentially expressed genes. Red indicates down-regulated genes; blue indicates up-regulated genes. Y-axes: -log10(Adjusted p-value), X-axes: Log2 (fold change: ectopic *Lhx3/4* ANP vs. control). Superimposed tables list top up- or down-regulated genes and include the reported human homologs of each *C. robusta* gene [57]. Note that "KY21.Chr" is not included but applicable to all gene ID numbers listed in the middle columns of all tables.

Shaded-out hits indicate ambiguous representation. Underlying data can be found in S2 Appendix. **(C)** Plot of all genes. The top 10 up- and down-regulated genes are highlighted. **(D)** Plot of annotated *C. robusta* TF genes [57,62]. The top 10 up- and down-regulated genes are highlighted. Bolded text highlights genes with characterized roles in cardiopharyngeal development. **(E)** Plot of the top 1,000 highest-expressed control sample (ANP-lineage) genes. **(F)** Plot of neural tube closure genes (as characterized in [80]). Table lists all up- and down-regulated genes. **(G)** Plotted collection of putative migration and cellular adhesion genes [81,82]. Table lists all up- and down-regulated genes. Genes with potential roles in cellular adhesion displayed in blue, actin dynamics in red, cell polarity in yellow, and extracellular matrix in black. **(H)** Plot of genes expressed at above average levels in the CPP lineage at stage 21 (mid-tailbud), as characterized through extensive, stage-specific single-cell RNA-sequencing [70]. The top 10 down-regulated genes and the top 30 up-regulated genes are highlighted. Bolded text highlights genes with characterized roles in migratory behavior. ANP, anterior neural plate; CPP, cardiopharyngeal progenitor; TF, transcription factor.

amenable to ectopic activation in the ANP in comparison to cardiopharyngeal specification programs that are executed in mesendodermal lineage cells.

Along with assessing the potential for *Lhx3/4* misexpression to ectopically activate non-ANP regulatory circuits, we also attempted to analyze the impact on ANP specification and morphogenetic programs. This involved plotting the top 1,000 most highly expressed genes from the control, wild-type ANP dataset (Fig 6E and S2 Appendix). This analysis indicated that *Lhx3/4* misexpression has a largely negative impact on ANP gene expression, significantly and robustly down-regulating 109 ANP genes versus only 10 that were up-regulated (Fig 6E). Furthermore, several of the top 10 most down-regulated genes have characterized roles in regulating ANP specification, including *Onecut* (KY21.Chr6.231) and *Six3/6* (KY21.Chr 10.262) or represent key downstream target genes such as *Tyrp-1* (KY21.Chr 5.389) [48,83,84]. Expression of a number of other characterized ANP transcription and signaling factors were also significantly down-regulated including *TCF/Lef* (KY21.Chr6.59), *Gsx* (KY21.Chr2.996), *Neurogenin* (KY21.Chr6.434), and *Pou4f* (KY21.Chr2.456) [48,62,85,86]. Thus, it appears that *Lhx3/4* misexpression suppresses execution of ANP specification and morphogenetic programs.

To better understand how *Lhx3/4* misexpression disrupted ANP morphogenesis (Fig 5), we also analyzed the expression of genes with potential roles in neural tube closure, actin dynamics, adhesion, cell polarity, and other morphogenetic processes. In line with the observed impact of *Lhx3/4* misexpression on neural tube invagination (Fig 5E and 5F), 10 of 61 genes with characterized roles in neural tube closure [80] were robustly and significantly down-regulated, while only one was up-regulated (Fig 6F). To explore other aberrant behaviors exhibited by *Lhx3/4*-expressing ANP lineage cells, we also analyzed the expression of genes that might contribute to cell adhesion or migration (Fig 6G). This included a list of 53 *C. robusta* orthologs to genes involved in cell polarity or actin dynamics [82], as well as a comprehensive set of 57 annotated orthologs to cadherin, protocadherin, laminin, and claudin genes ([81]; S2 Appendix). In line with the observed impact of *Lhx3/4* misexpression on the organization and coherence of the neural plate (Fig 5F), we detected down-regulation of genes that may encode proteins that contribute to epithelial adhesion and organization, including orthologs to *Cadherin18*, a protocadherin, a laminin, and *Prickle2*, a key planar cell polarity protein (Fig 6G).

As mentioned above, it does not appear that *Lhx3/4* misexpression was sufficient to drive full execution of the CPP specification program and downstream expression of *Foxf* target genes involved in CPP migration, such as *RhoD/F*. To further explore ectopic CPP gene expression, we extracted the gene expression profile of mid-tailbud stage CPPs from a single-cell embryonic database of *C. robusta* gene expression [70]. We chose this stage because it represents CPP gene expression 4 hours after the initial FGF-dependent induction, mirroring the 4-hour interval between *Lhx3/4* misexpression/FGF induction in the ANP and the RNA sequencing time point in our comparative analysis. Additionally, we reasoned that CPPs have begun their ventral migration at this time point (Fig 1D) and likely express genes involved in this behavior. We calculated the average normalized read count (0.44) for genes expressed in

the CPP lineage during the mid-tailbud stage (stage 21; [70]) and evaluated the impact of ectopic *Lhx3/4* on their expression levels in the ANP (Fig 6H and S2 Appendix). Strikingly, the majority of impacted CPP genes, 83 in total, were down-regulated, further supporting abbreviated execution of the CPP program. However, this analysis did reveal 53 up-regulated CPP genes including several orthologs to genes that have characterized roles in migratory behavior, including *FHOD3* [87], *GLIPR2* [88], *PLEC* [89], and *CTNNAL1* [90] (bold lettering, Fig 6H). Additionally, orthologs to some of the genes that were most highly up-regulated by *Lhx3/4* misexpression are also involved in cell migration (Fig 6C). For instance, the most highly up-regulated gene in this dataset is orthologous to *EPHA3*, an Ephrin receptor with documented roles in cardiac cell migration and in the de-adhesion of epithelial cells [91,92]. Furthermore, the second most highly up-regulated gene, *THSD7A*, has a role in actin cytoskeleton rearrangement and can induce filopodial extensions [93,94]. Thus, up-regulation of these CPP and non-CPP genes might contribute to the aberrant migratory behaviors observed in *Lhx3/4*-expressing ANP cells.

Collectively, comparative RNA sequencing analysis of *Lhx3/4*-expressing ANP cells revealed severe disruption of the native ANP program along with partial, highly abbreviated execution of the CPP specification program. While these results support the cofactor-dependent induction model, they do not indicate that lineage-determining factors function as "master-regulators" of differential induction downstream of a shared signal. Instead, these factors appear to participate as key players in complex, cascading circuits that dictate lineage-specific responses.

## Discussion

### A dual role for *Lhx3/4* in cardiopharyngeal progenitor specification

Previous studies have shown that CPP cells are specified through the combined activity of FGF-activated *Ets1/2.b* and an inferred ATTA-binding transcriptional cofactor. Here, our loss and gain of function data strongly support the hypothesis that *Lhx3/4* serves as this cofactor. Interestingly, this represents a second role for *Lhx3/4* in CPP specification, as it also contributes to founder cell specification earlier in development (Fig 1A). The resulting CPP specification circuit (Fig 3E) resembles an extended coherent feed-forward loop (FFL) [5]. In a typical FFL, a primary TF (A) participates in direct regulation of a target gene (C) while also participating in indirect regulation through a transcriptional cascade whereby (A) also activates a secondary TF (B), which coactivates the target gene (C). If all of the interactions are positive, then the FFL is considered to be a Type 1 Coherent FFL (C1-FFL) and serves to generate a "sign-sensitive delay." During CPP specification, *Lhx3/4* acts in a similar manner to the primary TF in a coherent type 1 FFL, activating target genes directly and indirectly through a transcriptional cascade (Fig 3E). However, in this case, the transcriptional cascade is more complex, as it involves 2 separated cascading steps (A activates B1, B1 activates B2, then A and B2 activates C). Network motifs are largely derived from studies of single-celled organisms, and it was predicted that these motifs would be more complicated and extended in developmental GRNs for multicellular organisms [5]. Studies of these more elaborate extended or interlinked motifs remain limited, and the functions of such motifs have not been well characterized [95]. This extended FFL may generate a sign-sensitive delay, but further testing will be required to examine potential impacts of this motif on regulatory dynamics. Functionally, this extended cascade links autonomous specification (Fig 1A) to non-autonomous, signal-dependent subspecification (Fig 1C). It will be interesting to investigate how often similar motifs are deployed in developmental GRNs and whether this represents a recurring topological feature, leveraging persistent expression of a lineage-determining TF to drive initial signal-independent

specification along with subsequent signal-dependent subspecification. It will also be critical to further assess the exact contributions of *Ets1/2.b* and *Lhx3/4* in cardiopharyngeal specification and to determine whether these 2 factors directly interact or function in a cooperative or semi-independent manner. Additionally, it will be interesting to examine whether *Lhx3/4* (or other members of this gene family) play similar roles during vertebrate cardiopharyngeal specification. Although previous studies suggest that *Lhx2* participates in cardiac and craniofacial development in mouse embryos [96], further studies will be needed to assess potential evolutionary implications regarding cardiopharyngeal development within the shared tunicate/vertebrate ancestor.

## Cofactor-dependent induction as a novel gene regulatory network subcircuit

Current reviews propose that a limited suite of subcircuit topologies capture the key regulatory logic of cascading gene regulation during development [8,13,14,20,21]. However, the existing slate of theoretical GRN subcircuits do not encompass the regulatory logic entailed by cofactor-dependent inductive signaling. To illustrate this, we have adopted diagrams previously deployed to depict the range of subcircuits observed in the well-characterized sea urchin developmental GRN [22]. According to these diagrams, inductive signaling entails a subcircuit in which gene expression is activated when both a signal Z (FGF), and the associated signal-dependent TF Y (Ets) are present (Fig 7A). However, according to this subcircuit, the same signal would always lead to the same transcriptional response. In order to explain signal specificity, we have combined this inductive signaling subcircuit with the previously posited "AND logic" subcircuit in which 2 TFs (W and X) coregulate gene expression (Fig 7B; [22]). The resulting "cofactor-dependent induction" subcircuit (Fig 7C) illustrates the key role of lineage-determining cofactors (X) in spatially restricting gene activation in response to widespread expression of signals and associated signal-dependent TFs (Y+Z). This model also illustrates how expression of multiple, spatially distinct cofactors ($X_1$-$X_4$) mediates differential, lineage-specific expression (colored boxes) in response to a shared signal (Fig 7D). As highlighted in this study, *C. robusta* provides a valuable experimental platform for investigating how widespread use of a shared signaling pathway (FGF/MapK/Ets) is translated into lineage-specific gene expression through the deployment of spatially restricted cofactors (Figs 1G and 7D).

Inclusion of the cofactor-dependent induction subcircuit will be essential for productive integration of many key signaling pathways into theoretical models of regulatory networks. In recent overviews, 2 subcircuit categories center on signaling, community effect, and toggle switch subcircuits [13,14]. Community effect subcircuits involve signal-dependent feedback loops that can generate a broad, regional transcriptional response to a spatially or temporally restricted initiating signal. Toggle switches serve to explain the regulatory logic underlying a subset of widely deployed developmental signaling pathways including Wnt, Notch, and Hedgehog. In these 3 signaling pathways, signal-dependent TFs act as repressors until they are converted to activators by receipt of the appropriate signal. Neither of these categories encompass the logic underlying differential, lineage-specific responses to the other 7 major families of highly pleiotropic developmental signals (TGF-ß, RTK, JAK-STAT, Hippo, Nuclear Receptor, NFK-ß, and JNK). In these 7 pathways, signal-dependent TFs do not function as default repressors. Rather, signal-dependent alteration of the localization or function of corresponding TFs can either promote or suppress gene expression in a context-dependent manner. The logic underlying transcriptional regulation by these 7 major signaling pathways is captured by cofactor-dependent induction subcircuit (Fig 7C). Thus, integration of this subcircuit category will help to rectify inaccurate representation of signal-dependent regulatory logic in current

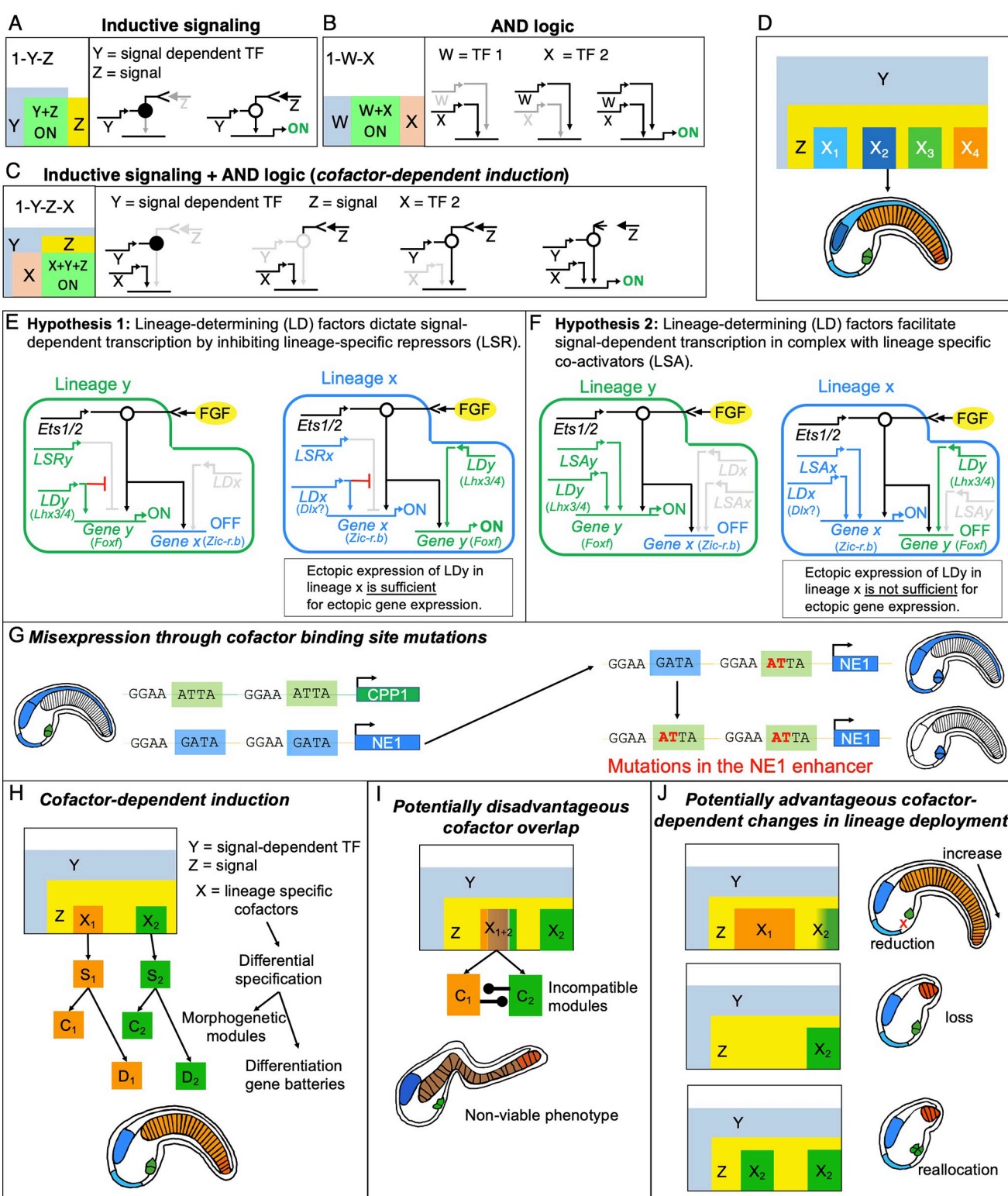

**Fig 7. Theoretical implications of cofactor-dependent induction. (A–C)** Subcircuit diagrams in which left hand boxes represent spatial domains for each contributing factor along with conditions that allow activation of each circuit, right hand boxes portray regulatory states, while gray lines and arrows indicate absence or inactivity of a particular factor. **(D)** Illustrations of spatially distinct outputs (color scheme from Fig 1G) due to differential expression of LD factors in accordance with the cofactor-dependent induction subcircuit (C). **(E, F)** Circuit diagrams illustrating 2 different models for the role of LD cofactors in signal-dependent transcription (see text). **(G)** Schematic illustrating the potential impact of LD factor binding site mutations on signal-

dependent expression of a hypothetical cardiopharyngeal progenitor lineage gene (CPP1, green) and neuroectodermal gene (NE1, blue). **(H)** Schematic illustrating the role of LD factors ($X_1$, $X_2$) in spatially restricted activation of specification modules ($S_1$, $S_2$) along with downstream morphogenetic modules ($C_1$, $C_2$) and differentiation gene batteries ($D_1$, $D_2$). Colors in accordance with Fig 1G. Note that these cartoons include 2 characterized, distinct notochord domains, primary notochord (orange) and secondary distal notochord (dark orange). **(I)** Schematic illustrating the hypothesized impact that overlapping expression of LD factors would have on morphogenetic programs. Cartoon tailbud depicts resulting nonviable phenotype. **(J)** Schematic illustrating hypothetical scenarios by which misexpression of LD factors might lead to selectively advantageous phenotypes. Top: primary notochord cofactor domain expansion generates more notochord cells or CPP cofactor domain restriction generates reduced cardiopharyngeal cells; center: the loss of a primary notochord cofactor results in a shortened tail, note that the distal secondary notochord (dark orange) is known to be independently specified; bottom: extended expression of the CPP cofactor in place of the primary notochord cofactor reallocates primary notochord lineage cells to increase the numbers of CPPs. CPP, cardiopharyngeal progenitor; LD, lineage-determining; LSA, lineage-specific coactivator; LSR, lineage-specific repressor; TF, transcription factor.

theoretical models. For example, signal-dependent TFs, including Ets family members, are often included in regulatory circuits without inclusion of the upstream signal mediating their activity or consideration of the logic mediating lineage-specific outputs [1,10,13,14,97–102].

## Potential roles for lineage-specific cofactors in signal-dependent transcription

Our identification of *Lhx3/4* as the lineage-determining factor for CPP specification extends the list of characterized partners mediating lineage-specific response to FGF, as illustrated in Fig 1G. In looking over this list, it becomes apparent that a simple model in which a single lineage-determining partner dictates differential, signal-dependent transcription is incomplete. In particular, some of these partners are in the same gene family. For example, the presumptive lineage-determining partners for CPP and ANP progenitors, *Lhx3/4* and *Dlx.b*, are both homeobox-family factors and would thus bind the same core nucleotides (ATTA). Therefore, it is not clear how enhancers would encode differential transcriptional responses to *Ets1/2.b* and *Lhx3/4* in versus *Ets1/2.b* and *Dlx.b*. One straightforward way to bypass this issue is to invoke the role of lineage-specific cofactors that also bind to signal-dependent regulatory elements (Fig 7E and 7F). These could either be lineage-specific repressors (Fig 7E, LSR) or lineage-specific coactivators (Fig 7F, LSA). Future studies will focus on characterizing these presumptive lineage-specific cofactors and their respective binding sites through informatic and wet-bench analysis of FGF-dependent regulatory elements in *C. robusta*.

## A primary, instructive role for lineage-determining factors in signal-dependent transcription

Current models of signal-dependent transcription do not indicate whether lineage-determining factors instruct or facilitate lineage-specific responses to pleiotropic signals [22,23,103,104]. The precise contribution of lineage-determining factors may be determined by their interactions with additional cofactors. For example, lineage-determining factors could instruct signal-dependent transcription by blocking a lineage-specific repressor (Fig 7E). Alternatively, lineage-determining factors could form a complex with one or more lineage-specific co-activators, facilitating signal-dependent transcription only when other members of the complex were available (Fig 7F). Based on this hypothetical framework, our results strongly suggest that lineage-determining factors play a primary, instructive role in signal-dependent transcription by blocking repressor activity (Fig 7E). In particular, we have shown that ectopic expression of *Lhx3/4* is sufficient for activation of CPP gene expression in the ANP (Figs 4E–4O, 6D and S4) as predicted by the "lineage-specific repressor" model (Fig 7E). The contribution of lineage specific repressors also serves to explain why ectopic CPP reporter expression extends into all 4 rows of the ANP including the anterior rows in which FGF-dependent expression of non-palp ANP genes are usually suppressed (Fig 4F and 4L). In particular, these

results suggest that CPP regulatory elements do not contain binding sites for repressors that modulate ANP expression in the palps. Further studies will be required to explore the role of presumptive lineage-specific repressors or cofactors and generate a precise model for the genomic encoding of signal-dependent transcription.

### Capabilities and limitations of lineage-determining transcription factors

Our data indicate that the roles of lineage-determining cofactors in inductive circuits may vary for different target genes. In particular, we found that misexpression of the CPP lineage-determining factor (*Lhx3/4*) drove ectopic expression of one CPP target gene (*Foxf*) in a purely FGF/MapK-dependent fashion while another target gene (*Hand.r*) displayed ectopic expression independent of FGF/MapK signaling (Fig 4). While the response of *Foxf* to this manipulation can be explained by invoking lineage-specific repressors (Fig 7E), the response of *Hand.r* appears to reflect an alternative regulatory logic in which high levels of cofactor expression (transgene-driven *Lhx3/4*) may overcome the need for a signal-dependent input.

Our data also highlight a key role for the regulatory landscape in modulating the impact of cofactor-dependent inductive circuits on downstream specification and morphogenetic programs. Based on our model, we predicted that ectopic expression of the CPP lineage-determining factor (*Lhx3/4*) would be sufficient to launch the CPP specification program along with downstream morphogenetic modules. We assumed that simultaneous execution of the native ANP program led to incompatible gene expression patterns that drove observed aberrations in ANP cell behaviors. However, our RNA-sequencing results did not align with this interpretation. Instead, these results indicated that ectopic execution of the CPP regulatory cascade was highly abbreviated, failing to fully extend beyond the expression of a subset of primary target genes (Fig 6H). These data also indicated that the native ANP regulatory cascade was severely disrupted leading to down-regulated expression of many key ANP TFs and downstream morphogenetic genes (Fig 6E and 6F). Intriguingly, it also appears that *Lhx3/4* misexpression drove ectopic activation of a MG patterning program, as reflected by robust up-regulation for a subset of TFs associated with this lineage. Thus, it appears that the capacity for lineage-determining factors to dictate lineage-specific inductive cascades is limited and context dependent. Although misexpression of a cofactor can launch an ectopic inductive program, further execution is likely constrained by the preexisting regulatory state of the target lineage. Further studies will be required to delineate how execution of conflicting inductive programs play out in different linages and at different time points in development.

### Implications of cofactor-dependent induction for disease states and evolution

Fully characterizing the regulatory logic underlying signal-dependent transcription will be critical for understanding and predicting the roles of nucleotide polymorphisms in disease ontogeny and evolutionary diversification. The regulatory elements encoding differential responses to highly pleiotropic signals may be particularly vulnerable to erroneous execution of specification or morphogenetic programs. This hypothesis is supported by previous studies regarding the use of suboptimal Ets binding sites in *C. robusta* regulatory elements encoding responses to FGF-dependent induction [105–107]. In these studies, optimization of Ets binding sites in anterior neural or cardiopharyngeal lineage enhancers was sufficient for widespread misexpression in other lineages undergoing FGF-dependent specification. Thus, it is likely that single nucleotide polymorphisms in other signal-dependent TF binding sites or in binding sites for lineage-determining cofactors could alter gene expression, thereby compromising developmental or physiological genetic programs. For example, 2 base pair

mutations could convert core binding motifs for the characterized neuroectodermal (NE) lineage-determining factor (GATA) in a *C. robusta* NE enhancer into core binding motifs for the cardiopharyngeal lineage determinant *Lhx3/4* (ATTA; Fig 7G). According to our current model (Fig 7E), this change would be sufficient to drive ectopic cardiopharyngeal expression (as illustrated in Fig 7G) assuming that NE enhancers lack binding sites for cardiopharyngeal lineage repressors (Fig 7E). A similar logic applies to potential misexpression of lineage-determining factors. As mentioned above, our *Lhx3/4* misexpression results (Figs 4 and 6) indicate that lineage-determining factors play a primary, instructive role in differential execution of specification programs in response to widespread signals. According to current models, the resulting execution of lineage-specific specification programs lead to downstream execution of lineage-specific morphogenetic and differentiation modules (Fig 7E and 7H). Thus, polymorphisms that result in overlapping expression of lineage-determining factors (Fig 7G) may result in execution of incompatible morphogenetic programs and contribute to the onset of inappropriate cell behaviors associated with congenital defects or disease (Fig 7I). From an evolutionary perspective, cofactor-dependent induction may represent a highly versatile GRN subcircuit subject to modifications that could result in selectively advantageous phenotypic changes. For instance, shifts in the expression domain of a lineage-determining factor could drive trait diversification through reduction, loss, or reallocation of a cell lineage (Fig 7J). Comparative analysis between the developmental GRNs in *C. robusta* and other diverse tunicates represent a potent framework for investigating whether these hypothesized regulatory shifts in cofactor-dependent induction subcircuits have contributed to trait loss, such as the loss of cardiopharyngeal structures in larvaceans [108–110] or trait diversification, as exhibited within highly divergent pelagic tunicate taxa [111].

## Materials and methods

### Molecular cloning

*Mesp>H2B::GFP* was previously described in [35]. *U6>sgRNA(F+E)* and *Mesp>nls::Cas9::nls* were generously provided by Lionel Christiaen [56]. Targeting *U6>sgRNA* plasmids were created by ligating annealed, phosphorylated oligo pairs into a BsaI-digested, empty *U6>sgRNA (F+E)* vector (S1 Table). *FoxfΔepi>LacZ* was created by fusing together 2 sections of DNA upstream of the *Foxf* gene. This approximately 200 bp deletion eliminates expression in the epidermis [39]. The first fragment spanned from 1,238 bp to 857 bp upstream of the *Foxf* TSS and was amplified using the 5′ oligo GAATGTGGTGTTATACTT and 3′ oligo CTCTCCAACGCGATCCAT. The second fragment spanned from 648 bp upstream of the TSS to 360 bp downstream of the TSS using the 5′ oligo CGAGTCGTAAAGCTTGCCG and 3′ oligo GCCTGATGTAATCTGTCTGC. This second fragment includes the *Foxf* 5′ UTR and the first 99 amino acids of the *Foxf* coding sequence, which was fused in frame with *LacZ*. *Hand.r>LacZ* was described in [39] (*Hand-like -1972-1793:296:lacZ*). *Dmrt>LacZ* and *Dmrt>GFP* were generated as in [48]. *Dmrt cis*-regulatory DNA was amplified with the 5′ oligo TCAGAACGAGGCGCTACATG and 3′ oligo TCACTGTTCTAAGCAAGGTATC AAGG [48]. *Dmrt>Lhx3/4* was generated by amplifying full length *Lhx3/4* (KY21.Chr13.457. v1.ND1-1) from mixed stage embryo cDNA using the 5′ oligo ATGATTCTCGATACTAAG GCGC and 3′ oligo CCACGTGACACATTTCCAA. This fragment was subcloned into the *Dmrt>GFP* vector after restriction digest removal of the *GFP* sequence. The sgLhx3/4-protected rescue construct (*Mesp>Lhx3/4^rescue^*) was created by removing *H2B::GFP* from our *Mesp>H2B::GFP* construct and inserting the full length *Lhx3/4* (KY21.Chr13.457.v1.ND1-1) sequence. This construct (*Mesp>Lhx3/4*) was then amplified with the oligos agcag**tt**ag-**ga**actggggacgtaggtCAATTCTATTTGCTGGATG and

cactacaatcgtcatctacatattg**CATGAAACATCTGAAGCATTC** (bold underlined nucleotides highlight induced mutations that do not affect translated amino acid sequence). About 1.0 μL of PCR product was used with the NEB Kinase-Ligase-DpnI enzyme mix (M0554S) per manufacturer's instructions. The resulting clones were screened via Sanger sequencing for presence of desired mutations.

## Electroporations

Gravid *C. robusta* were provided by MREP, collected from various locations near San Diego, CA, and kept under constant illumination in a recirculating refrigerated tank to promote the accumulation of gametes. Transgene DNAs in water used for electroporations were pooled together and mixed to a final concentration 0.77 M D-Mannitol. Specific amounts of transgene DNAs used per experiment were listed in S1 Table. Electroporations were carried out essentially as in [64], and electroporated embryos were left to develop at 18°C in 0.22 μm-filtered artificial seawater supplemented with penicillin/streptomycin (10 U/mL and 10 μg/mL, respectively). An individual trial refers to an independent electroporation on a different day using different parent animals.

## Sequencing of CRISPR-Cas9-induced indels

*Lhx3/4* CRISPR-Cas morphants coexpressing a *Mesp>H2B::GFP* reporter were developed until the early tailbud stage. Embryos were dissociated in magnesium and calcium-free seawater in 0.2% trypsin (Sigma Aldrich) on ice as in [36]. Trypsin was quenched using 1% BSA (Sigma Aldrich) and placed in calcium and magnesium-free seawater on ice. Dissociated GFP-positive cells were FACS collected using a BD Influx cell sorter (Penn Genomic and Sequencing Core, University of Pennsylvania). Captured cells were utilized directly as PCR template and amplicons containing the target sequences were generated using the 5′ oligo GTAGAC AATGCAGACCGGA and the 3′ oligo CGTCACAATACCTTGTATTCGTTGATCG. Amplicons were cloned into an empty vector and submitted for Sanger sequencing.

## Juveniles

After coelectroporation with *Mesp>nls::Cas9::nls*, *PC2>Kaede* (CITRES, https://marinebio.nbrp.jp/ciona/), and sgRNA plasmids targeting *GFP* or *Lhx3/4*, animals were kept on gelatin coated dishes for 20 to 22 hours at 18°C until reaching the swimming larval stage. Larva were transferred to gelatin-coated dishes with scratch marks that facilitate adhesion of the papillae to the dish necessary for metamorphosis. Animals were reared until reaching the end of the first ascidian stage (Stage 6: Days 7 to 9 postfertilization). Animals were first sorted based on the presence of the coelectroporation fluorescent marker (*PC2>Kaede*) and then heart morphology and peristalsis were assessed under transmitted light. Animals were scored as either normal or abnormal. Representative transmitted light videos were taken with a Zeiss LSM 980 confocal microscope. Those specimens were then relaxed using a menthol crystal placed in seawater until animals were unresponsive (approximately 15 minutes). Animals were fixed in 4% paraformaldehyde for 15 minutes at room temperature and stored at 4°C overnight. Animals were washed 3 times in 1X phosphate-buffered saline (PBS) for 5 minutes. Animals were then stained with 1:1,000 DAPI (Thermo Fisher Scientific) for 30 minutes at room temperature and washed in 1X PBS. Animals were placed in an optical dish and imaged using a Zeiss LSM 980 confocal microscope at 40× magnification in 1X PBS.

## LacZ staining

Embryos were fixed in MEM-GA (0.1 M MOPS (pH 7.4), 0.5 M NaCl, 2.0 mM MgSO4, 1.0 mM EGTA (pH 8.0), 0.2% glutaraldehyde, 0.05% Tween 20) for 30 minutes at room temperature, then washed 4X in 1X PBS plus 0.2% Tween 20 (PBST). Embryos were stained in a solution of 1% X-gal, 100 mM MgCl, 0.75 M Na2HPO4, 1.0 M NaH2PO4, 100 mM potassium ferrocyanide, 100 mM potassium ferricyanide, and 0.10% Tween 80, at room temperature for 2 to 4 hours, then washed 2X in PBST before transfer into 70% glycerol for imaging.

## Immunohistochemistry

Immunohistochemistry was executed as in [112]. Briefly, embryos were fixed in 2% paraformaldehyde, washed in PBST, and briefly permeabilized in ice-cold 100% methanol. Embryos were incubated at room temperature for 1 to 2 hours or 4˚C overnight with primary antibody (rabbit-anti-GFP, Thermo Fisher Scientific, A11122) at 1:1,000, washed 3 times in PBST, and incubated at room temperature for 1 to 2 hours or 4˚C overnight in secondary antibody (Alexa Fluor 488, A-11001, Thermo Fisher Scientific) at 1:500. Embryos were washed twice in PBST, incubated for 10 minutes in 50% glycerol/PBST and 1:1,000 DAPI, and then transferred to 80% glycerol/PBST for imaging.

## In situ hybridization

Whole mount in situ hybridization was performed essentially as in [113] with slight modifications. After several hours of staining, embryos were transferred out of staining buffer into 1X Tris-buffered saline and rocked at 4˚C overnight. The next day, embryos were transferred back into staining buffer for further colorimetric development. Upon completion, embryos were washed in PBST, fixed briefly in 4% paraformaldehyde, washed 2X in PBST, and transferred to 70% glycerol for imaging. Embryos probed for *Ets1/2.b* expression were transferred into 100% EtOH for approximately 5 minutes, then washed twice in PBST, prior to transfer into 70% glycerol.

## Imaging

Images of LacZ-stained and in situ hybridized embryos were captured through a 10X objective using a Nikon Alphaphot with an AmScope 10MP USB 2.0 Color CMOS Camera. Images and time-lapse movies of embryonic and larval fluorescence were captured on a Zeiss LSM 980 confocal microscope at 20× magnification. Fixed samples were imaged in 1X PBS, and live embryos were imaged in artificial seawater. S4–S8 Movies were generated by assembling time-lapse series of Z-stacks of live embryos from Stage 16 through Stage 25 using ImageJ software [114].

## mRNA sequencing and analysis

PolyA+ RNA was isolated from GFP+ cells (dissociated and FACS-collected as above; Penn Genomic and Sequencing Core, University of Pennsylvania). Paired-end libraries were prepared from >10 ng RNA (Takara SMART-Seq HT). Libraries were pooled and then sequenced on a lane of a NovaSeq 6000 S4 (Illumina) at Azenta Life Sciences, generating maximum reads of 150 nucleotides and a read depth of 45 to 62 million reads/sample (assessed using FastQC). Reads were then aligned by HiSat2 [115] to the ci3 assembly (UCSC Genome Browser). The *C. robusta* KH2012 gene models were used [116]. Referenced KY21 genes were manually converted using the KY genome browser [57]. KH IDs were input into the genome browser search bar. Results on the genome browser contained the KY IDs and homolog information. HTSeq

was used to perform gene counts [117]. DESeq2 was used to normalize gene count and calculate Log2 Fold Change and p-adjusted statistical significance [118]. A minimum of 100 reads was the threshold required for gene expression.

## Supporting information

**S1 Table.** **(Top)** Mass of transgene DNA used in each experiment. **(Bottom)** Sense strand sequences of sgRNAs used in this study.
(XLSX)

**S1 Fig. Full homeobox screen data. (Left)** Average percent of embryos displaying perturbed migration covering 2 trials for all of the sgRNAs tested in this study (* = $p < .05$). **(Right)** Table of *p*-values comparing sgRNA effects on migration. Column 1) *p*-values for negative controls vs. other sgRNAs; column 2) *Ets1/2.b* sgRNAs vs. other sgRNAs; and column 3) the 185/886 pair of *Lhx3/4* sgRNAs vs. other sgRNAs.
(TIF)

**S2 Fig. Detailed *Lhx3/4* targeting strategy. (A)** Diagram of *Lhx3/4* locus depicting the 2 isoforms of *Lhx3/4* and the locations targeted by *Lhx3/4* sgRNAs. **(B)** Sequencing results. WT spans upstream of the sgLhx3/4$^{185}$ position to downstream of the sgLhx3/4$^{886}$ position, (6,221 bp) indicates the nucleotides in the intervening wild-type sequence, underlining marks sgRNA NGGs. C06, C08, C14: sequenced clones of CRISPR morphant amplicons. Dashes indicate missing nucleotides in the sgRNA-targeted sequences. Two clones (C06/C14) had deletions, while one clone (C08) had a deletion and an addition of *TATA* (nonbolded).
(TIF)

**S3 Fig. Complete representation of the *Lhx3/4* rescue experiment. (A)** Left: representative embryos, inset numbers refer to quantity of embryos displaying the phenotype presented by each image over total GFP+ embryos for each group. Arrows point to normal CPP position at this time point. Green, GFP fluorescence. Right: graph of percent migration phenotype for each group for GFP+ embryos. Blocked migration refers to CPPs in the tail; excess migration refers to CPF-lineage anterior tail muscle cells in the trunk. $n = 300$ per condition. **** indicates *t* test *p*-value of $< .0001$. **(B)** Graph of overall percent GFP-negative embryos per condition.
(TIF)

**S4 Fig. Confocal imaging of *Lhx3/4*-activated *Hand.r>LacZ* in the ANP lineage. (Top)** Representative control embryo electroporated with *Hand.r>LacZ* and *Dmrt>GFP*. **(Bottom)** Representative embryo electroporated with *Hand.r>LacZ*, *Dmrt>GFP*, and *Dmrt>Lhx3/4* displaying *Hand.r>LacZ* reporter expression in the ANP lineage (white arrowheads). Confocal max projected Z-stacks were scored over 2 trials. In the *Dmrt* lineage, 13/13 (100%) of control embryos exhibited no *LacZ* staining, whereas 15/16 (93.7%) of experimental *Hand.r>LacZ* and *Dmrt>Lhx3/4* transgenic embryos were positive for *LacZ* staining in the *Dmrt* lineage.
(TIF)

**S1 Movie. Control juvenile showing peristalsis normally observed in the range of control animals (corresponds to Fig 3I).**
(MOV)

**S2 Movie. Experimental juvenile with sgRNAs targeting *Lhx3/4* that has no discernable myocardial cells (corresponds to Fig 3J).**
(MOV)

**S3 Movie. Experimental juvenile with sgRNAs targeting *Lhx3/4* that exhibits abnormal peristalsis due to severe disruption of myocardial tube morphogenesis.** Note that the heart appears to be compressed against the stomach. All videos contain juveniles that are in the first ascidian stage (Stage 6: Days 7–9).
(MOV)

**S4 Movie. Representative control embryo (*Dmrt>GFP*) displaying normal development and GFP expression in the anterior neural plate lineage.** S4–S8 Movies are constructed from confocal Z-stacks starting at Stages 15–16 (approximately 7 hours postfertilization) through Stage 25 at 20-minute intervals. Developing embryos are dorsal side up and anterior to the left.
(MOV)

**S5 Movie. A second representative control embryo displaying normal development of GFP-labeled cells.**
(MOV)

**S6 Movie. Representative experimental embryo (*Dmrt>Lhx3/4*) displaying aberrant behavior of GFP-labeled cells (described in text).**
(MOV)

**S7 Movie. A second representative experimental embryo displaying aberrant behavior of GFP-labeled cells.**
(MOV)

**S8 Movie. A third representative experimental embryo displaying aberrant behavior of GFP-labeled cells.**
(MOV)

**S1 Appendix. Raw data for Figs 2–5.** Raw counts for all of the experiments conducted in Figs 2–5. Tab labels refer to data presented in particular graphs.
(XLSX)

**S2 Appendix. RNA-sequencing data. (Tab 1)** DESeq2 statistics for all genes. **(Tab 2–6)** Lists of up- and down-regulated genes from Fig 6. First column lists the KH gene IDs for all genes that were selected for a particular analysis shown in each figure panel. The KH gene IDs for all relevant genes that were significantly up- or down-regulated are shown in blue or red font. Boxes indicate the most highly impact subset of genes as shown in the Fig 6 tables. For these genes KH IDs were paired with KY IDs and human homologs. **(Tab 7)** Referenced *Foxf* target genes.
(XLSX)

## Author Contributions

**Conceptualization:** C. J. Pickett, Bradley Davidson.

**Data curation:** C. J. Pickett.

**Formal analysis:** C. J. Pickett, Bradley Davidson.

**Funding acquisition:** Bradley Davidson.

**Investigation:** C. J. Pickett, Hannah N. Gruner.

**Methodology:** C. J. Pickett, Bradley Davidson.

**Project administration:** Bradley Davidson.

**Supervision:** Bradley Davidson.

**Visualization:** C. J. Pickett.

**Writing – original draft:** C. J. Pickett, Hannah N. Gruner, Bradley Davidson.

**Writing – review & editing:** C. J. Pickett, Hannah N. Gruner, Bradley Davidson.

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
