## [Editor Report · Decision Letter 0]

19 May 2023

Dear Dr Davidson, 

Thank you for submitting your manuscript entitled "Lhx3/4 acts as a cardiopharyngeal lineage determinant, dictating lineage-specific transcription in response to widespread fibroblast growth factor signaling" for consideration as a Research Article by PLOS Biology.

Your manuscript has now been evaluated by the PLOS Biology editorial staff as well as by an academic editor with relevant expertise and I am writing to let you know that we would like to send your submission out for external peer review.

Once your full submission is complete, your paper will undergo a series of checks in preparation for peer review. After your manuscript has passed the checks it will be sent out for review. To provide the metadata for your submission, please Login to Editorial Manager (https://www.editorialmanager.com/pbiology) within two working days, i.e. by May 23 2023 11:59PM.

Kind regards,

Richard

Richard Hodge, 

Associate Editor

PLOS Biology

rhodge@plos.org

---

## [Decision Letter · Decision Letter 1]

30 Jun 2023

Dear Dr Davidson,

Thank you for your patience while your manuscript "Lhx3/4 acts as a cardiopharyngeal lineage determinant, dictating lineage-specific transcription in response to widespread fibroblast growth factor signaling" was peer-reviewed at PLOS Biology. Please accept my apologies for the delays that you have experienced during the peer review process. Your manuscript has now been evaluated by the PLOS Biology editors, an Academic Editor with relevant expertise, and by three independent reviewers. 

In light of the reviews, which you will find at the end of this email, we would like to invite you to revise the work to thoroughly address the reviewers' reports.

As you will see below, the reviewers are generally positive and think the manuscript is interesting and well done. However, they raise some overlapping concerns, including the need to more directly demonstrate a combinatorial effect and FGF dependence of Lhx3/4 function, as well as strength of the reporting and genotyping of the CRISPR-based experiments. 

Given the extent of revision needed, we cannot make a decision about publication until we have seen the revised manuscript and your response to the reviewers' comments. Your revised manuscript is likely to be sent for further evaluation by all or a subset of the reviewers.

We expect to receive your revised manuscript within 4 months. Please email us (plosbiology@plos.org) if you have any questions or concerns, or would like to request an extension. 

**IMPORTANT - SUBMITTING YOUR REVISION**

*Re-submission Checklist*

*Published Peer Review*

*PLOS Data Policy*

*Blot and Gel Data Policy*

Sincerely,

Richard

Richard Hodge, PhD

rhodge@plos.org

REVIEWS:

Reviewer #1: The study by Pickett et al 'Lhx3/4 acts as a cardiopharyngeal lineage determinant, dictating lineage-specific transcription in response to widespread fibroblast growth factor signaling' proposes that a previously uncategorized subcircuit, co-factor dependent induction, is an important element of gene regulatory networks. More specifically, using the ascidian Ciona robusta, they show that the transcription factor Lhx3/4 acts as a co-factor of FGF-dependent cardiopharyngeal fate determination (cardiopharyngeal progenitor CPP induction during neurulation) downstream of its previously demonstrated function as a cell autonomous determinant of the same lineage earlier in development (cardiopharyngeal founder cell specification during gastrulation). Using tissue-specific CRISPR/Cas9 gene inactivation, they show that CPP identity and migration is impaired. Overexpression of Lhx3/4 in the anterior neural plate lineage, that is also specified in an FGF-dependent manner, is sufficient to activate the cardiopharyngeal genetic program and impairs neural tissue morphogenesis and differentiation.

This study, the underlying subcircuit and the theoretically implications are of great importance and potentially very interesting for a large readership. However, the primary data that are presented here suffer from a number of caveats that prevent a robust support of the model.

Major points:

1) The manuscript contains a number of mistakes and imprecisions (listed in the minor points) that make the reading difficult and question the thoroughness of the study.

2) General considerations on CRISPR/Cas9 gene experiments. There are a number of details to clarify regarding the method, the evaluation of the mutagenesis and whether mutations produce loss-of-function.

 - What is the control condition? Empty vector (line 187, Fig 2) or a sgRNA against GFP (line 228, Fig 3)? What is the quantity of plasmid that is electroporated?

 - What are the actual sgRNAs that have been used and lead to a phenotype (see minor points)?

 - A 16% indel frequency is shown in Fig 2F. However, since pairs of sgRNAs have been used, one expects to obtain deletion of the piece of chromosome between the 2 sgRNAs or indels at both sites. This is not shown.

 - Since the pair of efficient sgRNAs is unclear, one is left to imagine. From my own search, there are 2 possible scenarios. In the first case, 1 sgRNA targets the 5'UTR region and the other the coding sequence. The PCR designed to amplify the 5'UTR is thus irrelevant. In the second case, both sgRNAs target the 5'UTR. The PCR and sequencing are relevant but from the data, it seems that only one sgRNA is efficient, and this should be explicit. In addition, the authors should explain how, in this specific case, 5'UTR mutations would correspond to a loss-of-function.

 - For all targeted genes, the genomic locus (with gene/transcript/ORF) and sgRNAs positions should be displayed. Also, the mutation efficiency (and loss-of-function) should be evaluated.

3) A difficult issue is to distinguish the early and late functions of Lhx3/4. The CRISPR-mediated LOF is targeting the late FGF-dependent function. There are 2 main supporting evidence provided by the authors: the use of the Mesp enhancer and the lack of effect on Ets1/2.b expression. It would be important to further support the fact that early function is not impaired.

I suggest the following experiments. The authors should make sure that Mesp expression is not affected: either by in situ hybridization of the gene itself or by scoring the proportion of GFP positive cells or embryos in the experiments shown in Fig 2. The authors should also provide a rescue for the experiment in Fig 2. Finally, by expressing Cas9 using an earlier driver, it should be possible to block the early function of Lhx3/4, and the phenotype should be different (loss of Mesp progenitors).

4) Results in Fig 4 are exciting but should be expanded to robustly support the model proposed by the authors.

In Fig 4I it seems that the FoxF reporter is expressed in 3 rows of cells of the neural plate suggesting that it may also be expressed in the palp precursors. Since this territory is supposed to have silent FGF signaling (Wagner et al, 2012), the results do not fully match the model. I would thus recommend to accurately determine the reporter expression within the neural plate. It would also be interesting to determine FoxF (or other cardiopharyngeal genes) expression directly by in situ hybridization.

Next, it is essential to show that launching the cardiopharyngeal program is FGF dependent. The authors could perform the same experiment as in Fig 4I in embryos where FGF signaling is inhibited at the adequate stage. The authors could also express Lhx3/4 in the epidermis where FGF signaling is not active, and test whether activating the FGF pathway triggers a different response.

5) To further support the relevance of their model, it would be important to provide another example (not only the ectopic overexpression of Lhx3/4). I would suggest to drive Gata.a or Zic-r.b expression in the cardiopharyngeal lineage (using Mesp driver?) and determine the FGF-dependent consequences.

6) The authors have focused on cell behavior and morphogenesis following Lhx3/4 overexpression. It would also be interesting to look at the neural plate genetic program.

Minor points:

- Ets1/2.b being a direct target of Mesp is a critical element for the model and for the data in Fig 3F-H. Could the authors describe the supporting evidence?

- line 107: Ciona intestinalis b is uncommon. Should be Ciona intestinalis type B. A reference on the two types is missing.

- line 200: 'this pair'. Which one is it?

- Figure 2F: indicate sgRNA position and name.

- Figure referencing:

 line 129: should be 1C.

 line 299: 3O does not exist.

 lines 385-386: should not be 6C and 6D,E.

 supplemental table 1: it is not clear which experiments/figures the upper part of the table refers to.

- Unpublished/not shown data:

 line 125

 line 472

 line 643

- Inconsistencies between Supplemental Table 1 and Supplemental Fig 1:

 sgRNAs against Ets1/2.b do not have the same names.

 sgRNA LMX-478 is not listed in the table

Which sgRNAs have been actually used?

- Supplemental Fig 1. Why statistical tests have been conducted with the sgRNA combination against Lhx3/4 that has the weakest effect?

- Fig 3H: the legend should be % Ets1/2.b expression.

- Could the authors comment on the higher penetrance of the Lhx3/4 CRISPR phenotype in juveniles compared to embryos?

Reviewer #2: Pickett et al have investigated the role of Lhx3/4 in development of the cardiopharyngeal lineage in Ciona. The results of this clearly written and carefully documented study suggest that Lhx3/4 plays an important lineage determining role in cardiopharygneal cell migration and heart development. The authors argue that this is consistent with a role as a partner of FGF signal dependent Ets transcription factors, addressing the important question of how gene regulatory networks integrate extrinsic and intrinsic inputs. However, the following points appear necessary to reinforce the authors conclusions.

1. The authors convincingly show that Lhx3/4 is required for CPM migration, Hand gene expression and heart morphogenesis, and phenocopies Ets1/2 knock-down. These results are consistent with the authors' underlying hypothesis. However, in my reading of this manuscript it seems that these experiments do not directly demonstrate that Lhx3/4 is an Ets cofactor. Or that the role of Lhx3/4 is to confer lineage specific transcription in response to FGF signaling. As FGF signaling is also required for these phenotypes it is clearly challenging to show the FGF-signal dependence of Lhx3/4 function. Could the authors demonstrate molecular or genetic interactions between Ets1/2 and Lhx3/4 that would reinforce their interpretation? Can they rule out that Lhx3/4 plays FGF-independent roles in regulating cardiopharyngeal lineage migration? This should at a minumum be discussed and in the absence of such evidence the conclusions should be tempered.

2. This point also applies to the very interesting ectopic expression experiments that confer cardiopharyngeal migratory phenotypes and disrupt neuroectoderm development. Can the authors show that this is really through ectopic partnering with Ets factors as opposed to FGF-independent roles of Lhx3/4 in driving a cardiopharyngeal program? For example, can the authors show the Lhx3/4 phenotype is dependent on Ets1/2 or FGF signaling in the neurectoderm? Or could they show that ectopic Lhx3/4 activation of the Foxf or Hand enhancers is dependent on the Ets1/2 binding sites? In the absence of such evidence it might be more prudent to present the FGF signal partner role as one possible mechanism of Lhx3/4 function. 

3. Do the authors know if Lhx3/4 is involved in iterative roles at each step of CPM FGF signal response, through to activation of the pharyngeal muscle lineage? Are pharyngeal muscles also perturbed after Lhx3/4 inactivation? Discussion of the established role of Lhx2 in mammalian cardiopharyngeal mesoderm would be pertinent (PMID: 23112163).

4. The introduction and discussion could be significantly shortened. Some of the GRN points would be more suitable for a review, including parts or all of Figure 6, which is somewhat speculative, especially given points 1 and 2 above. If the Lhx3/4 role can be shown to depend on coincident FGF signaling this may be more relevant.

Reviewer #3: The paper nicely shows that Lhx3/4 is an instructive co-factor for FGF/MAPK-induced Ets transcription factor (TF) in the Ciona cardiopharyngeal (CP) lineage. This is convincingly shown with proper experiments, adapted controls and sufficient statistical evluation.

The originality lies in the conceptualization into a more broadly applicable gene regulatory subcircuit that takes into consideration the prerequisit of the signaling pathway activation of the TF (Ets) in addition to its mere presence. Such Ets pathway activation is then alteratively combined with different lineage determining TFs for tissue specific effector gene activation. While such combinatorial code was previuosly shown for other lineages the experimental proof of ectopic instructive behaviour and consequent 'confused' cellular morphology pinpoints this concept. In addition, Lhx assures a cascadal feed-forward loop in the CP lineage through two subsequent steps in synergistic activity with different co-factors whereby the later co-factor (Ets) is a consequence of Lhx early function (Lhx activating Mesp transcription that causes Ets transcription).

The conceptualization of a general inducible TF (Ets) activation principle in combination with different lineage determinants is important to a wider scientific community, also outside Ciona, notably because MAPK/Ets signaling is pleiotrophic and highly conserved. A differentially interpreted synergism with lineage determinats may generally explain how tissue specificity is achieved, also in the human cell differentiation and disease context.

Overall the paper is properly written, built and is adequately taking into account the literature etc. and can be recommended for publication - 

however, given that the following suggestions are taken into account.

The authors test a few candidate 'ATTA' binding TFs by CRISPR in the CP precursors (CPP) and these candidates were supposedly a result of their previous study. More precise support should be given why these TF are actual candidates (expression patterns etc.). 

Only Lhx CRISPR was successful but no indication was given that the guides/CRISPR for the other TFs actually worked. Should the authors have genotyping data for these other TFs it would strengthen the uniqueness of the Lhx effect. If not they may at least state that this is still an open question.

There are sections in the Results that should go in the Discussion as they interrupt the flow of data and don't actually contribute. This minimally concerns Figure 4B-D and associated text in lines 312 to 320 except for the rather important sentence of lines 317 to 318.

Furthermore, the Figure 6 should be compacted in A and B (and corresponding sections in Discussion lines 510 to 524) around the putative co-factor Dlx (that may also bind ATTA motifs) and should leave out consequent speculations about lineage specific activators and repressors. Either the authors bring the actual proof that Dlx functions this way and then speculate how the embryo sorts out two factors that both bind to ATTA motifs or they just leave that out since there may be another mechanism at work than speculated. This simplifies the message and streamlines it to what they have actually shown. 

More minor points and typos: 

Line 102: C. robusta is C. intestinalis Type A (not B)

Figure 2A: Please also include the older KH nomenclature, for easier comparison to previous studies

Line 360: change CFP to CPP

Figure 5C and E: DMRT>GFP would be an appropriate control, not Mesp>GFP 

Lines 381-382: Figure 5 not 6

Line 406: eliminate the braquet + text

Lines 436-446: correctly associate Figure 5 labels in the text and in the legend (H to L)

Line 629: may be BsaI not BsmBI

Supplement Table 1: correct to Figure 2A-F and Figure 3I-N (not 2)

---

## [Decision Letter · Decision Letter 2]

7 Dec 2023

Dear Dr Davidson,

Thank you for your patience while we considered your revised manuscript "Lhx3/4 acts as a cardiopharyngeal lineage determinant, dictating lineage-specific transcription in response to widespread fibroblast growth factor signaling" for publication as a Research Article at PLOS Biology. Please accept my apologies for the delays that you have experienced during this round of the peer review process. This revised version of your manuscript has been evaluated by the PLOS Biology editors, the Academic Editor and the original reviewers.

Based on the reviews, I am pleased to say that we are likely to accept this manuscript for publication, provided you satisfactorily address the remaining points raised by the reviewers. This includes adding the additional reporting details requested by Reviewer's #1 and #3 and shortening the Introduction and discussion sections as suggested by Reviewer #2. 

In addition, I would be grateful if you could please make sure to address the following data and other policy-related requests that I have provided below (A-E):

(A) We would like to suggest the following modification to the title: 

“Lhx3/4 initiates a cardiopharyngeal-specific transcriptional program in response to widespread FGF signaling”

(B) You may be aware of the PLOS Data Policy, which requires that all data be made available without restriction: http://journals.plos.org/plosbiology/s/data-availability. For more information, please also see this editorial: http://dx.doi.org/10.1371/journal.pbio.1001797

Thank you for already providing the individual numerical values that underlie the summary data displayed in the figures (Appendix S1 file). However, it seems that the figures specified in this document do not match the figure panels that we would require underlying data for, which are as follows: 

Figure 2B, 2F, 3D, 3H, 3L, 4G, 4J, 4O, 5D, 5K, 6B, S1, S3A-B

We would be grateful if you could double check the Appendix S1 file and edit the tabs if needed. In addition, we note that the Appendix S1 file does not include underlying data for the supplementary files (S1 and S3A-B) and we ask that this is included at this stage.

(C) Thank you for uploading the mRNA-sequencing datasets in the Appendix S2 file. Given the potential size of the dataset, we would encourage you to deposit this data in the GEO repository and provide the accession number in the Data Availability Statement of the online submission form. 

(D) Please also ensure that each of the relevant figure legends in your manuscript include information on *WHERE THE UNDERLYING DATA CAN BE FOUND*, and ensure your supplemental data file/s has a legend.

(E) Please ensure that your Data Statement in the submission system accurately describes where your data can be found and is in final format, as it will be published as written there. 

We expect to receive your revised manuscript within two weeks. 

*Published Peer Review History*

*Press*

Kind regards,

Richard

Richard Hodge, PhD

rhodge@plos.org

Reviewer remarks:

Reviewer #1: I am happy with the revised version of the manuscript that I find of much better quality.

There are some minor points that deserve corrections/clarifications:

- lines 236-238: should be Figure S2

- Table S1 is missing from the submission

- Fig S2B:

 The sequence of sgLhx3/4(185) does not correspond to the sequence present in the 1st submission and to the displayed sequence of the mutant clones (1 SNP).

 The sgLhx3/4(886) size/PAM do not fit either.

 The displayed mutant clones seem different from the ones shown in Fig 2F of the 1st submission.

 The frequency of mutant clones is not indicated.

- The RNAseq data should be deposited in a public repository, and the accession number provided in the manuscript.

- RNAseq data were analyzed using KH2012 gene models, but KY21 gene models are shown in Fig 6. Please indicate how the conversion was made.

Reviewer #2: The authors have extensively revised their nice manuscript in the light of the earlier comments. This includes addition of important experiments demonstrating that ecoptic Lhx3/4 fails to induce the CPP program in the neural plate when the MAPK pathway is blocked. This study brings a number of new findings and directions with respect to how regulatory inputs are integrated during development. The introduction and discussion remain long and could potentially be shortened.

Reviewer #3: Most of my concerns were adressed/explained except for the following: 

1. I could not find Supplement Table 1. It is important to list the sequences of all guides for the different genes used in this study - notably as some of them were not extensively studied

2. The KH and KY21 nomenclature mix is still irritating - there is no match between Figure 6 and the corresponding lists in Appendix S2, this needs to be adjusted: please add the KY21 names and human orthologs in the Appendix S2 list as of Figure 6.

3. For Figure 6 the highlighting in bold and color coding (Fig.6G notably) needs to be clarified in the legend

4. Lines 236 and 238: Figure S4 and S5 should be Figure S2 A and B

5. Figure S3 B scoring of negative embryos is not meaningful - or explain why negative rather than positve embryos are scored (as requested by Reviewer 1 point 3); in the text (line 249) Figure S2 should be changed to S3

---

## [Editor Report · Decision Letter 3]

21 Dec 2023

Dear Dr Davidson,

On behalf of my colleagues and the Academic Editor, Anna Kicheva, I am pleased to say that we can accept your manuscript for publication, provided you address any remaining formatting and reporting issues. These will be detailed in an email you should receive within 2-3 business days from our colleagues in the journal operations team; no action is required from you until then. Please note that we will not be able to formally accept your manuscript and schedule it for publication until you have completed any requested changes.

PRESS

Best wishes, 

Richard

Richard Hodge, PhD

rhodge@plos.org

PLOS
